# Leveraging the water-environment-health nexus to characterize sustainable water purification solutions

Yu-Li Luo[1], Yi-Rong Pan[2,3], Xu Wang [1,4] ✉, Zhao-Yue Wang[1], Glen Daigger [5], Jia-Xin Ma[2], Lin-Hui Tang[1], Junxin Liu[2], Nan-Qi Ren[1] & David Butler [4]

Chemicals of emerging concern (CECs) pose critical threats to both public health and the environment, emphasizing the urgent need for effective water treatment measures. Yet, the implementation of such intervention technologies often results in increased energy consumption and adverse environmental consequences. Here, we employ a comprehensive methodology that integrates multiple datasets, assumptions, and calculations to assess the human health and environmental implications of removing various CECs from source water. Our analysis of two treatment alternatives reveals that the integration of riverbank filtration with reverse osmosis offers a promising solution, yielding healthier and more environmentally favorable outcomes than conventional sequential technologies. By incorporating context-specific practices, such as utilizing renewable energy sources and clean energy technologies, we can mitigate the adverse impacts associated with energy-intensive water treatment services. This research advances our understanding of the water-health-environment nexus and proposes strategies to align drinking water provision with public health and environmental sustainability objectives.

The advancement of chemistry greatly contributes to enhancing societal well-being over the past decades, as evidenced by a substantial increase in both the amount and variety of chemicals utilized. In regions with available data, such as the United States and Europe, it is estimated that the current market comprises ~75,000 to 140,000 chemicals[1]. Moreover, global chemical sales surpassed 5.6 trillion US dollars in 2017, with projections suggesting they may double by 2030 (ref. 2). This progress, however, has inadvertently resulted in the contamination of drinking water sources with complex mixtures of chemicals that exhibit spatial and temporal variability[3], posing important threats to public health and the environment[4]. Numerous chemicals of emerging concern (CECs), such as pesticides, industrial additives, pharmaceuticals, personal care products (PCPs), and disinfection byproducts (DBPs), have been increasingly detected in drinking water and blood serum samples[5,6]. Concerns about human exposure are escalating due to evidence that these chemicals can have deleterious biological effects at extremely low concentrations, ranging from nanograms to micrograms per liter[7,8].

The increasing variability and complexity of drinking water quality necessitate the adoption of a diverse array of water treatment technologies[9]. These technologies are often integrated into sophisticated process configurations designed to meet evolving challenges. Treatment processes typically begin with conventional methods such as aeration, rapid sand filtration, coagulation, sedimentation, and filtration, and progress to advanced techniques including carbon filtration, advanced oxidation, desalination, and membrane filtration[10]. As water treatment facilities are upgraded to meet new quality standards, these elaborate configurations become essential for achieving optimal

[1]State Key Laboratory of Urban Water Resource and Environment, School of Civil and Environmental Engineering, Harbin Institute of Technology, Shenzhen, Shenzhen, China. [2]Research Center for Eco-Environmental Sciences, Chinese Academy of Sciences, Beijing, China. [3]International Institute for Applied Systems Analysis, Laxenburg, Austria. [4]Centre for Water Systems, University of Exeter, Exeter, UK. [5]Department of Civil and Environmental Engineering, University of Michigan, Ann Arbor, MI, USA. ✉e-mail: wangxu2021@hit.edu.cn

treatment outcomes. However, the sustainability of these practices is often compromised by a heavy reliance on chemical additives, high energy consumption, substantial environmental emissions, and demanding operational requirements[11,12]. Furthermore, these plants usually require large, centralized infrastructure, substantial financial investment, specialized engineering expertise, and extensive treatment facilities[13,14]. Additionally, the intensive use of chemicals in water treatment can exacerbate contamination issues, notably through the formation of harmful DBPs such as trihalomethanes and haloacetic acids, which pose severe health risks and increase environmental pollution[15–17].

Nature-based solutions, such as riverbank filtration (RBF), offer multiple advantages for drinking water purification by leveraging natural processes like soil filtration and biological degradation[18,19]. These systems effectively remove contaminants while reducing the reliance on energy- and chemical-intensive treatments, thereby enhancing sustainability and lowering operational costs[20]. Despite these benefits, the substantial presence of CECs in groundwater often necessitates additional treatment for potable use[21,22]. Reverse osmosis (RO), which is effective at removing a broad spectrum of CECs, has been combined with RBF in various pilot studies globally to evaluate its technical feasibility[23,24]. The combination of RBF and RO (RBF-RO) systems has shown potential in reducing chemical dosing and costs without compromising public health. However, the current RO membranes lack sufficient selectivity for target pollutants, leading to high energy consumption, especially when removing low-molecular-weight neutrally charged chemical pollutants (ranging from 100 to 200 Daltons)[25]. Achieving sustainability in these practices requires a comprehensive understanding of their broad systemic impacts, which remains limited.

Here, we integrate diverse datasets, assumptions, and calculations to model the cancer and non-cancer disease burdens associated with various CECs in source water. We explore the potential for reducing these burdens by integrating RBF with RO and conventional sequential technologies. Additionally, we evaluate the life cycle environmental impacts of different water purification systems, identify impact hotspots, and develop optimization strategies to mitigate adverse effects. Our previous research indicated that geographical, climatic, and other context-specific factors greatly influence these impacts[26]. Consequently, in the current study, we differentiate our primary results based on the specific data concerning the electricity generation mixes of 136 countries, examining whether environmentally favorable outcomes can be achieved by implementing energy-intensive RBF-RO systems in diverse global contexts. Building on these findings, we introduce a water-environment-health nexus (referred to as the WEALTH approach) that elucidates and coordinates interactions among drinking water supply, public health protection, and environmental sustainability goals.

## Results

### Overview of the water purification systems

We selected a drinking water production plant in Kamerik, Netherlands, for further modeling and analysis due to its utilization of two parallel water treatment processes. The plant provides $2.4 \times 10^6$ m³/y of drinking water characterized by low hardness and color intensity, chemical and biological stability, and priority substance concentrations that are half the maximum levels stipulated in Dutch regulations[27]. Specifically, the plant utilizes an extended treatment (ET) train, referred to as the RBF-ET system in this study, which comprises biological iron removal with dry filtration (including aeration and rapid sand filters), pellet softening, carry-over filtering, ion exchange, granular activated carbon, and ultraviolet light disinfection (Supplementary Fig. 1a). This system purifies water sourced from wells adjacent to the LeK River. Additionally, the plant employs an innovative treatment system combing RBF and RO, designated as the RBF-RO system. This system utilizes RO to filter the same source water, with subsequent post-treatments including ion exchange, remineralization,

oxygenation, and degasification to conform to drinking water standards and enhance water taste (Supplementary Fig. 1b). The essential process parameters of the two drinking water treatment systems are outlined in Supplementary Table 1. Comprehensive modeling efforts involved integrating various datasets and assumptions to evaluate the removal efficiency of 93 CECs, which include 41 pesticides, 19 industrial chemicals, 17 pharmaceuticals, 7 antibiotics, 5 DBPs, and 4 PCPs across the two water treatment systems (Supplementary Table 2). The selection of these CECs was based on their prevalence in scholarly literature, their relevance to water purification, and the availability of datasets suitable for modeling purposes. These CECs serve as representative contaminants to simulate the performance and assess the impacts of water purification systems, potentially indicating similar challenges posed by other waterborne contaminants. The analytical framework established for this study is further detailed in Supplementary Fig. 2, providing a schematic representation of the processes involved and the modeling methodologies applied. This comprehensive approach aims to enhance understanding of the efficacy of different water treatment technologies in mitigating the presence of diverse contaminants, thus contributing to the development of more effective and sustainable water purification strategies.

### Modeled disease burdens associated with CECs in each water production system

To assess the burdens of cancer and non-cancer disease associated with CECs present in drinking water from the alternative treatment systems, we quantified the disability-adjusted life years (DALYs) for annual exposure to the reference CECs, considering both carcinogenic and non-carcinogenic effects. The individual DALYs for each treatment system were calculated by aggregating the Monte Carlo (MC) iterations of annual DALYs for the specified exposure pathway. To evaluate whether each water treatment system pose a tolerable health risk, we employed an upper limit of $1.00 \times 10^{-6}$ DALYs person$^{-1}$ year$^{-1}$, a threshold established by the World Health Organization (WHO) for drinking water[28]. The cumulative probability distribution function (CPDF) of individual DALYs was utilized to provide insights into the median DALYs and the range of probable annual DALYs, reflecting the inherent variation in model parameters (Fig. 1a). Our analysis revealed that for source water, 0% of the MC simulations for cancer risks and 11% for non-cancer risks resulted in DALYs $\leq 1.00 \times 10^{-6}$ person$^{-1}$ year$^{-1}$, underscoring the critical need for water treatment for safe drinking purposes. For the RBF-ET system, 0% of the MC simulations for cancer risks and 100% for non-cancer risks meet the $\leq 1.00 \times 10^{-6}$ person$^{-1}$ year$^{-1}$ criterion, indicating effective mitigation of non-carcinogenic risks. Conversely, all MC simulations in the RBF-RO system resulted in disease burdens of $\leq 1.00 \times 10^{-6}$ DALYs person$^{-1}$ year$^{-1}$ for both cancer and non-cancer risks. Additionally, we evaluated the health benefits associated with transitioning from the RBF-ET system to the RBF-RO system by calculating the difference in MC simulations of annual DALYs for each treatment process (Fig. 1b). The results indicated positive health benefits for the RBF-RO system in 100% of MC simulations for cancer diseases and in 85% for non-cancer diseases (Fig. 1b), suggesting that switching to the RBF-RO system would likely be advantageous from a public health perspective. This finding supports the potential public health benefits of adopting more advanced water treatment technologies such as RBF-RO in mitigating the risks posed by CECs in drinking water.

### Role of system components in reducing the disease burden

In this study, we conducted a comprehensive analysis addressing both variability and uncertainty to evaluate the impact of three fundamental system components—RBF, ET, and RO—on reducing the burdens of cancer and non-cancer diseases associated with CECs present in source water. Figure 2 illustrates the relative contribution of different CEC categories to the changes in disease burdens following the

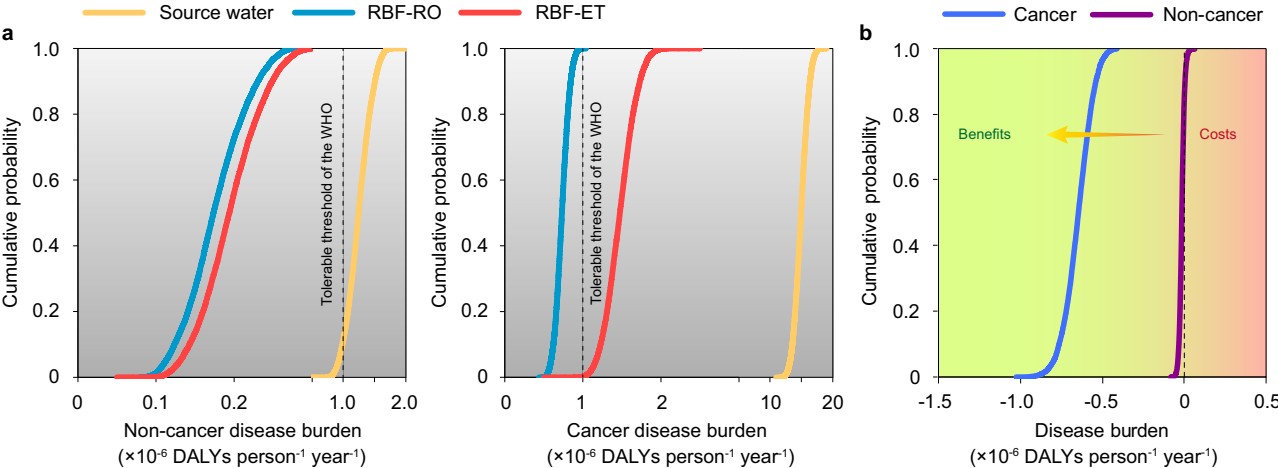

**Fig. 1 | Estimated non-cancer and cancer disease burdens associated with source water and drinking water treated via two alternative purification systems.** **a** Cumulative probability distribution function (CPDF) curves depicting the disability-adjusted life years (DALYs) associated with annual exposure to chemicals in source water (orange curve), drinking water treated by the combination of riverbank filtration and reserve osmosis (RBF-RO) system (blue curve), and drinking water purified by the integration of riverbank filtration and extended treatment (RBF-ET) system (red curve). The dotted vertical lines denote the upper limit of the tolerable disease burden ($1.00 \times 10^{-6}$ DALYs person$^{-1}$ year$^{-1}$) as suggested by the World Health Organization (WHO). **b** CPDF curves illustrating the human health effects of RBF-RO, determined by the total annual DALYs per person exposed. Negative values signify the health benefits acquired upon transitioning from RBF-ET to RBF-RO. The disease burdens were assessed in distinct carcinogenic and non-carcinogenic impact categories based on 10,000 Monte Carlo simulation runs.

enhancement of treatment infrastructure from a standalone RBF system to an integrated RBF-ET or RBF-RO systems. Notably, pesticides were identified as the primary category of CECs showing a great reduction in non-cancer disease burdens, as depicted in Fig. 2a. The median non-cancer burden decreased from $8.29 \times 10^{-7}$ DALYs person$^{-1}$ year$^{-1}$ in the source water to $4.19 \times 10^{-7}$ DALYs person$^{-1}$ year$^{-1}$ in the RBF effluent, further declining to $1.20 \times 10^{-7}$ DALYs person$^{-1}$ year$^{-1}$ and $1.35 \times 10^{-7}$ DALYs person$^{-1}$ year$^{-1}$ following the implementation of RO and ET, respectively (Supplementary Table 3). Conversely, DBPs were identified as the principal category of CECs demonstrating a decreasing trend in median cancer burdens, due to their higher initial concentrations, as illustrated in Fig. 2b. The median cancer burden initially decreased from $1.25 \times 10^{-5}$ DALYs person$^{-1}$ year$^{-1}$ in the source water to $4.67 \times 10^{-6}$ DALYs person$^{-1}$ year$^{-1}$ in the RBF effluent, subsequently reducing to $1.17 \times 10^{-6}$ DALYs person$^{-1}$ year$^{-1}$ and DALYs $5.96 \times 10^{-7}$ person$^{-1}$ year$^{-1}$ following the completion of ET and RO, respectively (Supplementary Table 3). Although all three system components contributed to a reduction in disease burdens associated with CECs in treated water, the RBF-RO configuration demonstrated superior performance, resulting in median cancer and non-cancer burdens of $6.84 \times 10^{-7}$ DALYs person$^{-1}$ year$^{-1}$ and $1.51 \times 10^{-7}$ DALYs person$^{-1}$ year$^{-1}$, respectively. This analysis underscores the effectiveness of integrating advanced treatment processes to achieve remarkable reductions in health risks associated with CECs in drinking water.

### Exposure concentrations, human toxicity factors, and disease burdens of CECs

Figure 3 illustrates the exposure (residual) concentration, human toxicity factor, and associated cancer burden of each CEC in drinking water treated via the RBF-RO system. Our analysis focused on the median cancer burden, as outlined in Supplementary Table 4, which provides comprehensive model outputs. This focus was guided by the analogous patterns observed in both cancer and non-cancer risks. Notably, the highest cancer disease burdens were correlated with elevated residual concentrations of CECs. For instance, treated water containing high levels of chloroform (a DBP) at a concentration of 129.2 ng L$^{-1}$, and naphthalene (an industrial chemical) at a concentration of 79.0 ng L$^{-1}$, exhibited relatively high disease burdens, calculated at $1.76 \times 10^{-8}$ DALYs person$^{-1}$ year$^{-1}$ and $5.40 \times 10^{-8}$ DALYs person$^{-1}$

year$^{-1}$, respectively. This indicates a direct relationship between the concentration of CECs in treated water and the resultant health risks. Furthermore, the analysis demonstrated that CECs with high toxicity factors importantly contributed to the cancer burden, even when their residual concentrations in the treated water were relatively low. For example, the residual concentration of nitrosodimethylamine (a DBP) was only 4.6 ng L$^{-1}$, similar to that of diethylhexyl phthalate (an industrial chemical) at 6.9 ng L$^{-1}$. However, the resulting cancer burden from nitrosodimethylamine was markedly higher at $3.39 \times 10^{-8}$ DALYs person$^{-1}$ year$^{-1}$, compared to a remarkably lower burden of $1.22 \times 10^{-10}$ DALYs person$^{-1}$ year$^{-1}$ from diethylhexyl phthalate. This disparity is primarily owning to the much higher toxicity factor of nitrosodimethylamine (11.9 cases kg$^{-1}$) compared to that of diethylhexyl phthalate ($2.9 \times 10^{-3}$ cases kg$^{-1}$), underscoring the importance of considering both concentration and toxicity in assessing the health impacts of CECs.

### Environmental impacts throughout the water treatment process

Our study initially focused on modeling the non-cancer and cancer disease burdens associated with CECs in drinking water produced by alternative systems. However, to fully evaluate the broad environmental impacts, we conducted a life-cycle assessment (LCA). Figure 4a illustrates that, compared to the RBF-ET system, employing the RBF-RO system for drinking water production resulted in important reductions in various environmental burdens. These reductions include terrestrial acidification (63% reduction), freshwater eutrophication (23%), mineral consumption (16%), fossil fuel depletion (14%), marine ecotoxicity (14%), freshwater ecotoxicity (12%), ozone depletion (5%), and global warming (1%). Nonetheless, the RBF-RO system exhibited ~53% higher marine eutrophication potential compared to the RBF-ET system. This discrepancy suggests that nitrogen-containing substances, possibly emitted during upstream membrane production and electricity consumption, greatly contribute to marine eutrophication. Notably, post-treatment processes in the RO system, such remineralization through the addition of calcium and magnesium, are necessary to comply with drinking water regulations and improve water taste. This remineralization typically involves mineral injection into drinking water using $CO_2$, which is often sourced from industrial

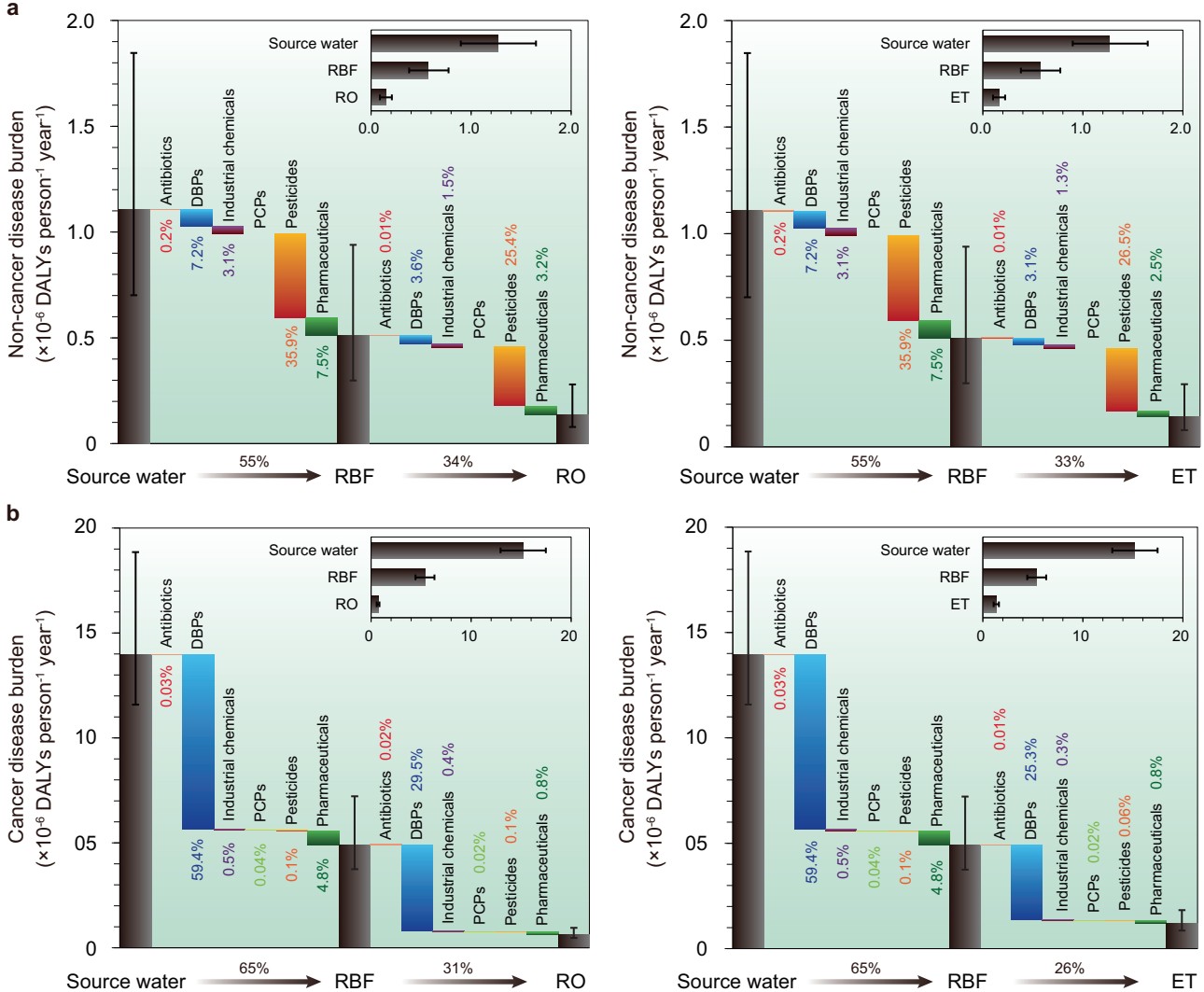

**Fig. 2 | Contribution of each category of CECs to the changes in disease burdens associated with drinking water after continuous treatment series. a** non-cancer disease burden. **b** cancer disease burden. Each column height represents the cumulative median contribution of each chemical of emerging concern (CEC) category, with error bars indicating the 5th and 95th percentiles based on 10,000 Monte Carlo simulations. Small histograms within each chart illustrate the estimate uncertainties, with column lengths showing mean values and bars representing 95% confidential intervals derived from the same simulations. RBF: riverbank filtration. RO: reserve osmosis. ET: extended treatment. DBPs: disinfection byproducts. PCPs: personal care products. DALYs: disability-adjusted life years.

processes that use monoethanolamide as an adsorbent[29]. However, the volatile nature of monoethanolamide and its degradation products can lead to contamination of terrestrial ecosystems through atmospheric deposition[30], explaining the higher terrestrial ecotoxicity potential observed with the RBF-RO system compared to the RBF-ET system. Moreover, our analysis revealed that the transportation of chemicals and materials from upstream factories to the drinking water production plant also contributes to higher terrestrial ecotoxicity. This is primarily attributed to the use of trucks, assumed to be the primary mode of transport in this study. Substantial contributors to terrestrial ecotoxicity include zinc, copper, and other heavy metals related from truck brakes, as detailed in Supplementary Table 5. These findings underscore the importance of considering a logistic environmental perspective when evaluating water treatment systems, accounting for the entire lifecycle from raw material extraction through end-use and potential disposal.

### Potential to minimize unwanted impacts
In Fig. 4a, we highlighted the important impacts of marine eutrophication attributable to the production of materials and electricity for

the RBF-RO system. To mitigate these adverse effects, we explored enhancements in material efficiency and the adoption of energy recovery measures. Initially, we posited that advancements in membrane science and engineering could double the lifespan of RO membranes[31], thereby reducing the volume of membranes requiring transportation. Additionally, we examined the potential of incorporating a Pelton turbine to facilitate energy recovery from the high-pressure concentrated brine. Notably, nearly 50% of the power consumed in the RBF-RO system is due to high-pressure pumps, as illustrated in Supplementary Fig. 3. Pelton turbines, known for their high efficiency and uncomplicated mechanical design, provide an efficient solution for hydraulic energy recovery. They operate with only the turbine back pressure required for shaft seal operation, eliminating the need for high inlet pressure[32]. As depicted in Fig. 4b, integrating strategies to extend the service life of RO membranes and to harness hydraulic energy from high-pressure brine could potentially reduce the marine eutrophication potential by ~24%. This integrated approach not only demonstrates effectiveness in mitigating terrestrial eutrophication but also shows potential in reducing terrestrial ecotoxicity and other environmental impacts associated with the RBF-RO system.

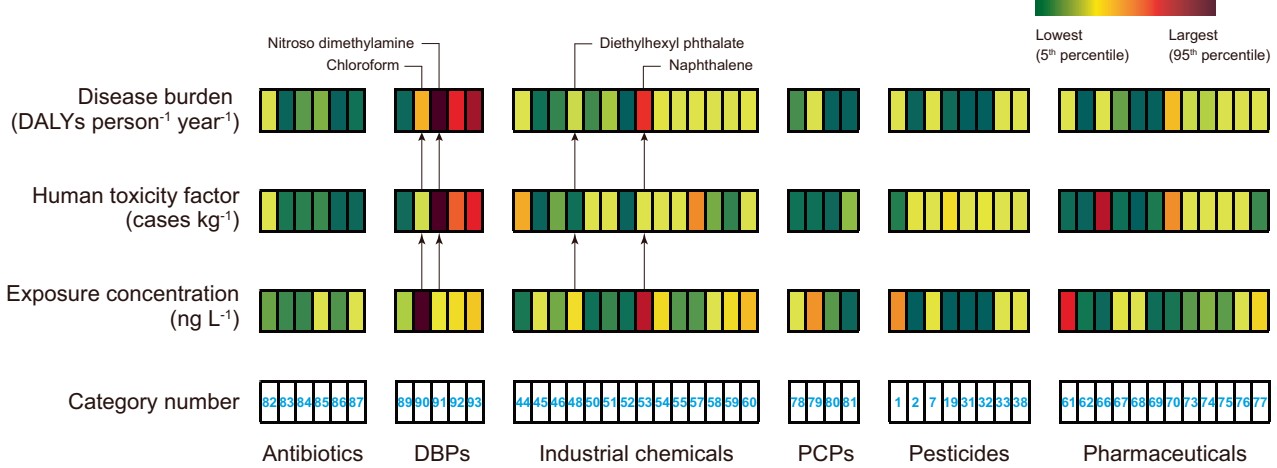

**Fig. 3 | Exposure concentrations, human toxicity factors, and cancer burdens of six categories of CECs in drinking water treated via RBF-RO.** The color scale denotes the magnitude of each exposure concentration, toxicity factor, and disease burden, while also highlighting the contributions of exposure concentrations and toxicity factors to the disease burden of each chemical of emerging concern (CEC) in drinking water produced by integration of riverbank filtration and reserve osmosis (RBF-RO). The charts are based on median data sets for each CEC. Note: CECs lacking available data on carcinogenic effects are not included. The chemical number of each CEC is provided in the Supplementary Information. DBPs: disinfection byproducts. PCPs: personal care products. DALYs: disability-adjusted life years.

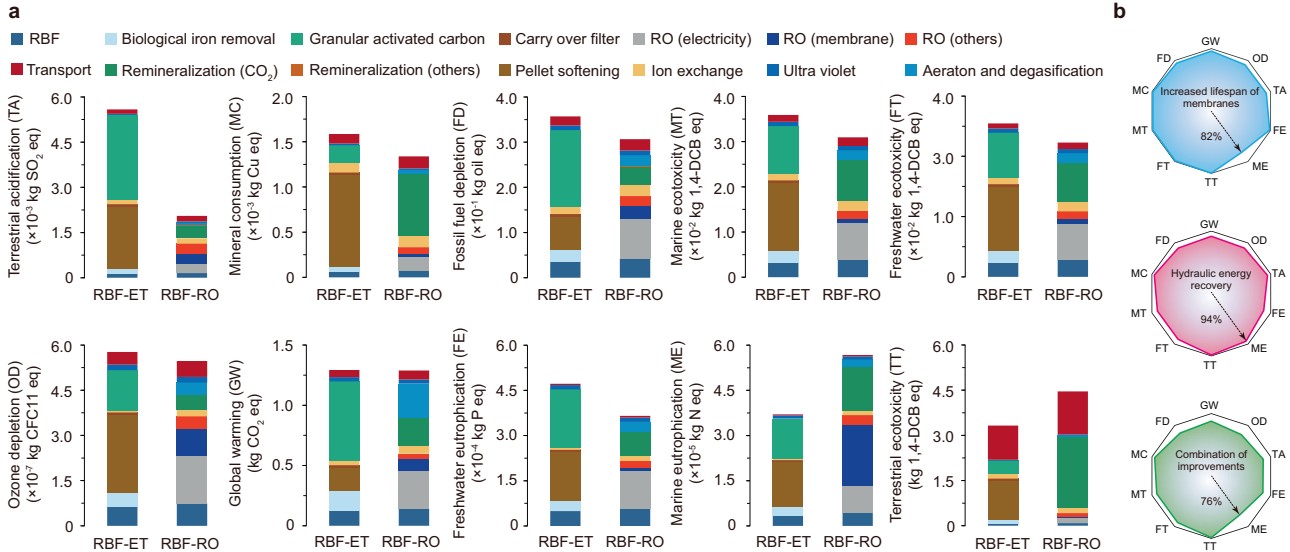

**Fig. 4 | Life-cycle environmental impacts of two alternative water treatment systems. a** Contributions of different water treatment processes to 10 mid-point environmental impacts, expressed per cubic meter of drinking water produced over 25 years of operating the combination of riverbank filtration and extended treatment (RBF-ET) as well as the integration of riverbank filtration and reserve osmosis (RBF-RO) systems. The relative size (or absence) of each color illustrates the contribution of the process to each environmental impact. **b** Performance and co-benefits of RO-related optimization strategies to mitigate marine eutrophication are depicted for the RBF-RO system. Additional details regarding other constituents associated with RO are provided in the Supplementary Information.

Furthermore, our analysis revealed that modifying the transportation logistics of chemicals and materials from trucks to more environmentally friendly conveyances, such as ships, could further diminish the terrestrial ecotoxicity potential of the RBF-RO system by ~9%, as detailed in Supplementary Fig. 4. This strategy represents a holistic approach to improving the environmental footprint of water treatment systems, focusing on both upstream material production and downstream operational efficiencies.

### Evaluation of RBF-RO in different world contexts
In previous research, the composition of electricity sources was shown to remarkably influence the environmental impacts of energy-intensive water treatment methods[33], with the efficiency and impacts of electricity generation processes varying greatly based on geographical factors[34]. Consequently, we conducted an assessment of the location-specific environmental consequences of the RBF-RO system for 136 countries across Africa, the Americas, Asia, Oceania, and Europe (enumerated in Supplementary Table 6), integrating country-specific data on electricity generation mixes (refer to Fig. 5a). The selection of these countries was guided by distinct primary energy compositions and electricity generation technologies capable of diversifying energy-related impacts[35]. Figure 5b illustrates maps summarizing the spatial divergence in the environmental repercussions of the RBF-RO system due to variations in the electricity mix across different geographic regions. Notably, regions like Canada and Brazil show effective mitigation of environmental impacts, which is evident

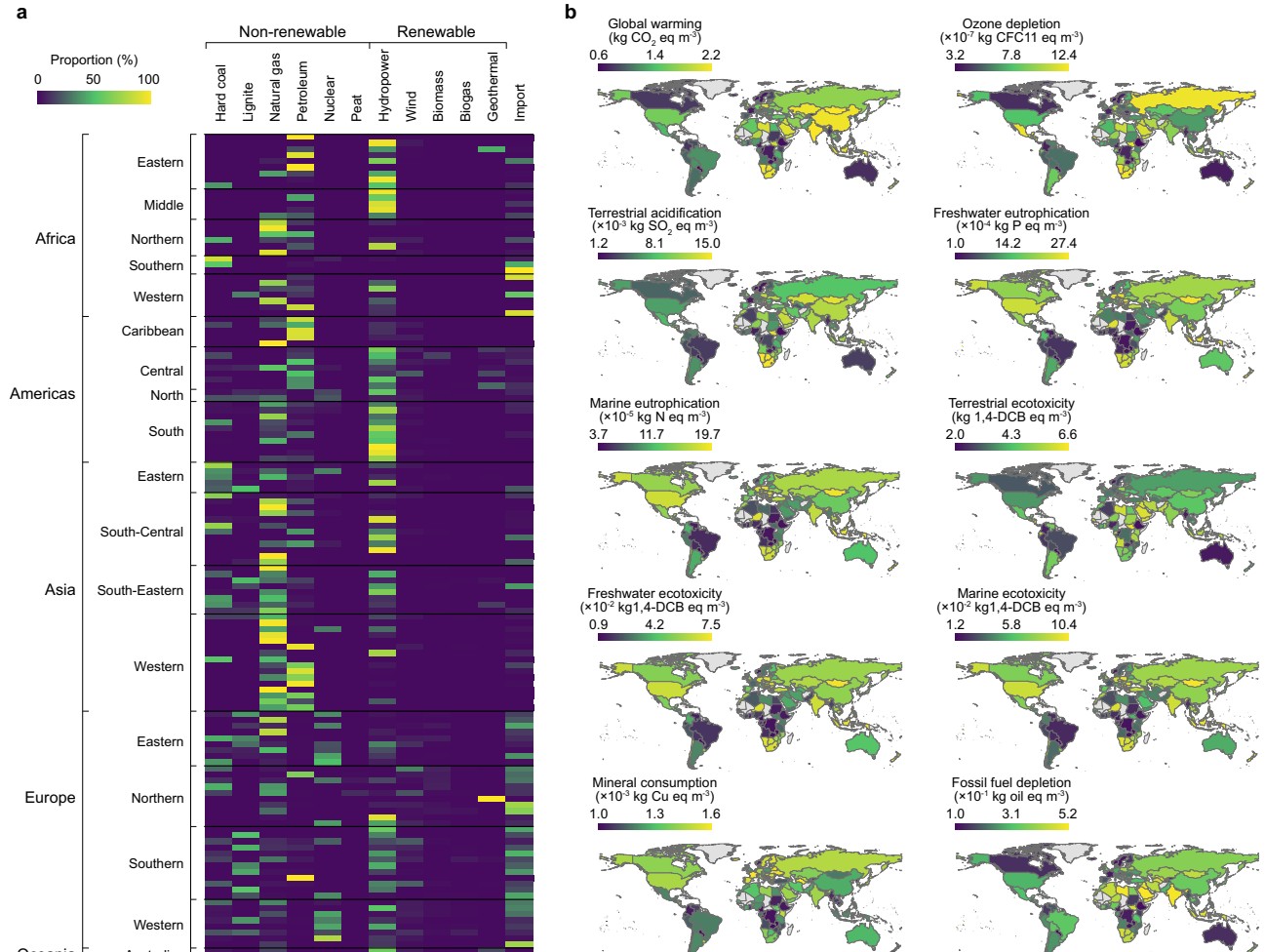

**Fig. 5 | Spatially differentiated electricity generation mix and resultant effects on the environmental performance of the RBF-RO system at a global scale.** **a** Variation in the electricity generation mix among 136 countries (listed in Supplementary Table 5). The color scale represents the relative proportion of energy sources in each country. **b** Maps displaying the selected 136 countries, with colors representing the environmental impacts of the riverbank filtration and reserve osmosis (RBF-RO) system implemented in each country. The global maps were generated using QGIS, and the country boundary data is sourced from http://www.naturalearthdata.com/. Supplementary Data 1 provides the complete model outputs.

from the dark shading across almost all impact maps. This indicates that the adoption of renewable resources, such as hydropower in electricity generation, can substantially mitigate the overall environmental repercussions of the RBF-RO system. Conversely, the electricity compositions of China and India are heavily reliant on hard coal, with a minimal share of renewable sources. Consequently, it is comprehensible that the implementation of an RBF-RO system in these countries would result in heightened impacts in terms of global warming, terrestrial acidification, and ecotoxicity. Despite China and India having similar electricity generation mixes, the deployment of an RBF-RO system in China would yield comparatively lower environmental impacts. This is partly attributed to the utilization of less environmentally damaging coal technologies in China, which optimize coal combustion and reduce emissions of $SO_2$ and other contaminants (see Supplementary Table 7). Less environmentally damaging coal technology encompasses various technologies and methodologies aimed at mitigating the environmental footprint of coal-based energy production and curbing emissions of greenhouse gases and particulate matter, a prominent example being the adoption of carbon capture and storage technologies[36]. Additionally. it was observed that lignite, a lower-grade coal variant, constitutes a great portion of Australia's electricity mix, leading to elevated eutrophication and ecotoxicity potentials in aquatic ecosystems. This is primarily attributed to the

considerably higher fugitive emissions of $CO_2$, $NO_x$, $SO_2$, and particulate matter from lignite power plants compared to those from the combustion of hard coal or natural gas (refer to Supplementary Table 8). These findings highlight the critical role of regional energy policies and the composition of power generation in shaping the environmental impacts of advanced water treatment measures like the RBF-RO system.

## Sensitivity analysis of disease burdens and environmental impacts

In this study, we conducted a comprehensive sensitivity analysis to evaluate the impact of various CECs on disease burden and environmental outcomes within the framework of water treatment processes. Our findings, illustrated in Supplementary Fig. 5, indicate that the disease burden is predominantly sensitive to specific CECs, notably naphthalene (categorized as an industrial chemical), nitroso dimethylamine (a DBP), and nitroso pyrrolidine (also a DBP), which were identified as primary contributors to cancer risks. Methamidophos, another industrial chemical, emerged as the most important CEC influencing non-cancer risks, exhibiting a Spearman's correlation coefficient of 0.90 for influent concentration and 0.25 and 0.26 for the removal rates of RBF and RO, respectively. It is noteworthy that both cancer and non-cancer disease burdens demonstrate a higher

sensitivity to influent concentration data compared to the removal rate data of the respective CECs, emphasizing the pivotal role of influent concentration in determining the overall disease burden. Furthermore, the sensitivity of removal rate data was observed to be influenced by the toxicity factor of the CECs. For example, despite similar influent concentration sensitivities for naphthalene and nitroso dimethylamine, a marked difference in the sensitivity of removal rate data was noted, with nitrosodimethylamine, characterized by a higher toxicity factor (11.9 cases kg$^{-1}$), exhibiting greater sensitivity than naphthalene (0.07 cases kg$^{-1}$).

When evaluating environmental impacts, illustrated in Supplementary Fig. 6, several process parameters were identified as important contributors to environmental outcomes. These include the recovery rate of the RO process, the lifespan of the membrane module, the energy efficiency of the pumps used in the RO unit, and the utilization of $CO_2$ for remineralization. The RO recovery rate, which represents the proportion of system inflow converted into product water, influences water extraction from the RBF process, associated transportation energy, electricity demand for high-pressure RO filtration, and the volume of RO brines requiring subsequent treatment. Consequently, environmental categories highly sensitive to energy consumption are disproportionately affected by variations in this parameter. Similar energy-related effects are observed in the efficiency of pump electricity in both RO and RBF units. The utilization of $CO_2$ in the remineralization process emerged as the second most sensitive parameter due to substantial upstream production impacts associated with it. The lifespan of the membrane module is another critical factor, as it determines the membrane requirement over the operational period of the RBF-RO scheme. Enhancing the lifespan of the membrane module could remarkably reduce the demand for membrane replacement and mitigate the associated environmental impacts of the production process. These findings underscore the complexity of optimizing water treatment processes, where adjustments in one component can greatly affect both health and environmental outcomes.

## Discussion

Water is indispensable for life and well-being, yet its quality is under severe threat, posing important public health risks[37]. Recent research suggests that ~16% of premature deaths worldwide can be attributed to environmental contamination[38], highlighting the critical need for interdisciplinary approaches[39]. These approaches should integrate analytical chemistry, exposomics, and water pollution control to elucidate the complex relationships between contaminant exposure, environmental quality, and health outcomes[40,41]. The issue is further complicated by the presence of CECs in drinking water, which often manifest at low concentrations and present analytical challenges due to their intricate chemical profiles and synergistic effects[42]. Additionally, long-term exposure to multiple CECs can exacerbates health hazards[8]. Our model-based analysis of two real-world treatment approaches suggests that the integration of RBF and RO systems can substantially enhance drinking water quality, maintaining disease burdens associated with CECs well below the WHO's tolerable limits. We suggest adopting a cause-effect framework, departing from traditional engineering approaches, to focus not only on residual pollutant concentrations but also on the potential disease burden posed by CECs with high toxicity factors at low exposure levels in drinking water. This perspective is supported by extant experimental literature[43]. Thus, it is vital to establish a framework for utilizing toxicity data to identify, prioritize, and address emerging chemical contaminants in future water purification research. For instance, DBPs, noted for their high toxicity in our studies, exhibited important toxicity reductions when treated using RO protocols, as opposed to the existing ET scheme shown in Fig. 2. Furthermore, addressing site-specific variability in contaminant composition—due to agricultural runoff, urbanization,

industrial discharges, and natural geological features—is essential for customizing treatment strategies and regulatory measures that effectively mitigate risks and protect public health against evolving water quality threats[14]. Moreover, the relevance of per- and polyfluoroalkyl substances (PFAS) to human health through drinking water exposure, despite their exclusion from this study due to data limitations, cannot be overstated. PFAS are known for their persistence and potential adverse health impacts, representing critical concerns in drinking water assessments[44]. Urgent targeted research is necessary to advance analytical methods for PFAS detection, perform comprehensive exposure assessments, and develop tailored risk assessment models to evaluate the health impacts of PFAS[45]. Bridging these knowledge gaps will augment our understanding of PFAS-related risks and facilitate the development of targeted mitigation strategies to protect public health from the effects of emerging contaminants in drinking water.

Our study, focusing on specific integrated systems such as RBF-RO, highlights the need for broader research into alternative water treatment methodologies. We encourage the scientific community to explore a wider array of treatment techniques, facilitating a thorough investigation of the synergies between natural ecological processes and engineered systems. Integrating natural methods such as biofiltration, phytoremediation, or wetland treatment with innovative engineered solutions could unveil novel strategies that optimize contaminant removal while minimizing energy use and environmental impact. Future research should evaluate the feasibility and scalability of merging natural and engineered processes across various environmental settings, aiming to develop holistic and sustainable water treatment strategies that effectively address emerging contaminants and safeguard long-term water resource health. Moreover, variations in solute characteristics (e.g., molecular size, charge) and membrane properties (e.g., pore size, material composition) greatly affect the efficacy of contaminant removal in different RO systems[46,47]. These differences, arising from variations in feedwater composition, operational conditions, or membrane configurations at treatment facilities, must be considered when applying research findings to other RO settings. Despite inherent challenges, our research provides valuable insights into the performance of a specific RO system for the removal of CECs, laying a foundational understanding of factors that influence treatment efficacy under controlled conditions. Future studies should delve into these complexities through advanced modeling techniques and experimental validation to clarify the intricate relationships between solute properties, membrane characteristics, and removal efficiencies in diverse RO contexts. Addressing these complexities will enhance the reliability and practical application of RO-based water treatment strategies, advancing the provision of safe and sustainable drinking water. Furthermore, the viability of implementing RBF-RO systems varies importantly based on regional characteristics such as climate, hydrogeology, and initial water quality[48,49]. For instance, in humid regions with plentiful surface water, RBF may serve as a reliable and cost-effective pre-treatment option for RO[50,51]. Conversely, in arid regions with limited surface water or challenging water qualities, direct RO from groundwater or desalination might be more suitable[52]. Adaptations in less favorable areas might include enhanced pre-treatment processes or the integration of advanced membrane technologies to meet specific water quality challenges.

Integrating RBF with RO for drinking water purification can simplify process configurations and reduce the reliance on treatment additives, thereby diminishing negative environmental impacts. Nonetheless, it is imperative to address potential drawbacks, such as marine eutrophication and terrestrial ecotoxicity. For example, our findings suggests that fugitive emissions of nitrogen-containing contaminants during the production of RO membranes may contribute to marine eutrophication. Similarly, the process of remineralizing RO water with minerals and $CO_2$ could increase terrestrial ecotoxicity due to $CO_2$ sequestration. Additionally, the post-treatment of RO brines

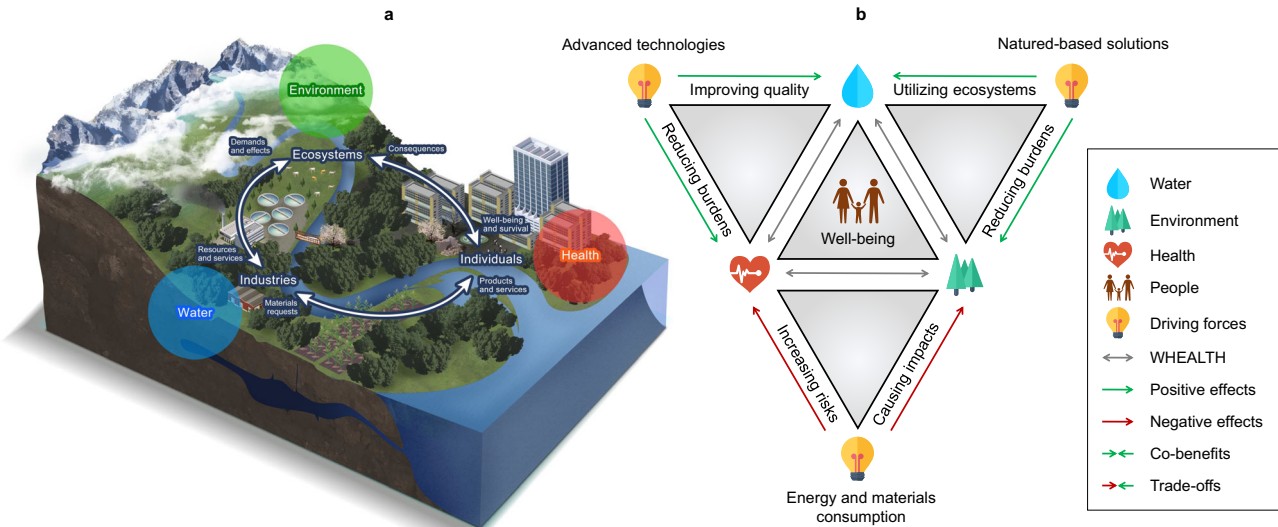

**Fig. 6 | Conceptual framework illustrating the interactions among clean water supply, human health protection, and environmental sustainability goals ("WEALTH"). a** Advancements in water purification technologies are critical for addressing increasing demands on water quality and securing access to clean and safe water. The development and integration of innovative technologies in water purification require rigorous piloting and testing, impacting various supporting industries, such as energy and chemical products. These sectors are essential in enhancing quality of life through rapid advancements in products and services. Within this framework, the environment plays a pivotal role by providing ecosystem services that are vital for a holistic approach to water treatment. Environmental outcomes can be either adversely or positively influenced by decisions made by individuals and industries, which in turn rely on natural resources. **b** The unfolding of **a** illustrates all bidirectional interactions within the WEALTH framework. displays all bidirectional interactions within the WEALTH framework. It exemplifies how the integration of advanced technologies with ecological processes, such as the RBF-RO system discussed in this study, can enhance the harmony among water quality, human health, and environmental outcomes.

can have important environmental repercussions. Furthermore, our research also highlights the substantial energy consumption associated with the operation of high-pressure pumps in RBF-RO systems, underscoring the urgency of pursuing optimization efforts. These efforts should include extending the lifespan of membrane modules, exploring green remineralization technologies, and implementing energy recovery strategies. Although RBF-RO systems can alleviate adverse health and environmental impacts, particularly in regions with access to renewable energy resources such as Africa and South America, the associated costs cannot be overlooked. Initially, the costs of upgrading, operating, and maintaining advanced treatment facilities such as RO-based systems may exceed those of existing systems that target known pollutants[42]. Financially constrained water treatment systems, particularly in developing countries, may find implementation of RO systems challenging despite its efficacy against multiple emerging pollutants. This situation highlights the risk of underestimating the true costs of conventional systems by neglecting emerging threats in drinking water sources. A comprehensive assessment of the costs and benefits of treatment options is crucial to safeguard human health and the environment[53]. Systems facing financial limitations might benefit from prioritizing known contaminants and adopting cost-effective water treatment practices. It is essential to weigh the long-term benefits and costs of various treatment alternatives to make informed decisions that ensure sustainable and effective water treatment solutions.

The imperative to enhance water treatment services is driven by technological advancements, public health considerations, industrial requirements, and the complexities of environmental stewardship[54,55]. Our study emphasizes the urgent need for the development of holistic methodologies that integrate the interactions among water, health, and the environment. This approach aims to identify potential trade-offs, co-benefits, and optimization strategies throughout the lifecycles of water production systems, as illustrated in Fig. 6. Although these areas are interconnected, global challenges such as ensuring clean water,

preserving ecosystems, and safeguarding public health have traditionally been addressed in isolation[56]. For example, while engineered measures like RO-based technologies effectively remove contaminants from drinking water, our research adopts a fresh perspective by exploring the broader implications of drinking water production. We highlight that the intensive electricity consumption and resultant greenhouse gas emissions from these processes can adversely affect both human and natural systems. In response, we introduce the water-environment-health nexus (WEALTH) approach. This methodology is designed to systematically reflect, model, and analyze the variables and drivers within the interconnected realms of drinking water supply, human health, and environmental impacts. By adopting the WEALTH approach, we aim to foster a deeper understanding of the interconnectedness of these systems and facilitate more sustainable and comprehensive solutions in water treatment practices. This approach not only seeks to improve water quality but also aims to reduce environmental footprints and enhance public health outcomes, demonstrating the importance of integrated and interdisciplinary solutions in addressing complex global issues.

To translate advanced environmental understanding into actionable, decision-ready information, predictive, system-scale, and robust modeling tools are crucial for rapid assessment. LCA provides a framework to meet these needs, yet it risks misrepresenting decision impacts if it fails to account for site-specific processes and spatial heterogeneity[57]. In our study, we found that the environmental effects of energy-intensive water production practices are greatly influenced by the local electricity mix. Nations with plentiful renewable energy resources and advanced power technologies can integrate energy-intensive purification technologies like RO more effectively[58]. Our model accounts for variations in the site-specific electricity generation mixes. However, changes in feed water quality, which also affect energy consumption[59], were beyond our study's scope. Addressing the

energy intensity of RO operations in response to diverse feed water conditions might involve optimizing system design, incorporating energy recovery devices, or deploying advanced control strategies to enhance efficiency. By considering regional and operational nuances, water treatment strategies can be tailored for optimal performance and sustainability across various geographical and hydrological contexts. Moreover, our study integrated sensitivity analyses to enhance the robustness and reliability of our findings. These analyses allow us to explore the impacts of varying parameters and assumptions on model outcomes. Systematically altering inputs, such as chemical concentrations or treatment efficiencies, helps identify key factors that influence results and determine the assessment conclusions' sensitivity to uncertainties. Insights from these sensitivity analyses are instrumental in guiding water treatment practices. They help prioritize crucial factors in the design and operation of purification systems, ensuring that environmental and health impacts are minimized. Understanding which treatment processes or chemicals have the most important effects on environmental outcomes informs decision-making for optimized strategies and effective resource allocation. Furthermore, sensitivity analyses support policy development by underscoring the sensitivity of environmental and health outcomes to regulatory parameters, thereby promoting evidence-based decision-making for sustainable water treatment services.

Our current modeling approach, while informative, may not fully encompass the dynamic nature of water quality changes within treatment systems. Since alterations in one unit process can greatly affect subsequent treatment steps[60], these changes introduce complexities that our simulations do not comprehensively represent. Future studies should consider employing more sophisticated modeling techniques that integrate actual process modeling approaches, incorporate real-time data monitoring, and implement feedback loops to improve accuracy and adaptability to system variations[61]. Furthermore, while our research provides a foundational framework for estimating the removal of CECs at the system level, it is evident that more nuanced models are necessary to capture the intricate interactions and dependencies between unit processes. The development of integrated models that account for the specific physicochemical properties of CECs, the dynamic nature of treatment processes, and potential synergistic or antagonistic effects among unit operations is crucial. This advancement will optimize treatment strategies and ensure the provision of safe drinking water amidst emerging chemical threats. Our study contributes valuable insights by offering a structured approach to estimate removal efficiencies based on empirical data and process parameters. Future research efforts should aim to develop these integrated models further, enhancing our ability to address the complexities of water treatment and ensuring the continued protection of water resources from emerging chemical contaminations. Moreover, in estimating health burdens associated with exposure to multiple CECs in drinking water, we employed a widely accepted additive approach in mixture toxicity modeling[62]. This approach assumes that CECs act independently, and their combined effects can be predicted by summing individual health burdens. This simplifies calculations, leveraging available comprehensive toxicity data for individual CECs, though it neglects potential interactions. We compared the additive approach with other mixture toxicity models (e.g., dominant, multiplicative) to evaluate the robustness and reliability of the health burden estimates obtained in this study. Future research should focus on addressing these uncertainties and refining mixture toxicity modeling techniques. This includes advancing our understanding of chemical interactions, exploring non-linear dose-response relationships, integrating more comprehensive exposure data, and incorporating variability in human susceptibility. Additionally, exploring the implications of mixture toxicity in real-world scenarios

and assessing risk management strategies based on different modeling approaches will translate research findings into actionable public health measures. This will enhance our ability to evaluate and mitigate the health impacts of emerging chemical contaminants in drinking water effectively.

The analyses presented in this study can be enhanced through the integration of additional site-specific data, which may address potential uncertainties, particularly concerning the assumptions made about the pollutant removal capacities of riverbank ecosystems. These capacities may vary regionally due to differences in soil and aquifer properties[49]. In our study, the pollutant removal factors were derived from existing literature, with the application of probabilistic distributions to accommodate site-specific uncertainties. We employed MC simulation to capture the variability among model parameters and to address uncertainties in the DALY estimates through confidential intervals. While the potential uncertainties were within acceptable ranges, we conducted additional sensitivity analyses to demonstrate how varying levels of parameter uncertainty could influence our results. Moreover, variations in industrial production and consumption patterns, as well as the release, fate, transport, and treatment of chemical pollutants, may differ substantially across different countries and regions[63]. Our research highlights the importance of conducting scenario analyses on a global scale to garner valuable insights and to account for site-specific practices and conditions. While our data stem from a comprehensive global literature review, emphasizing geographic relevance is crucial in addressing emerging chemicals in source water and treatment processes. Despite the regional variability of specific emerging chemicals, driven by diverse industrial activities, agricultural practices, and environmental conditions, the challenges and importance of addressing these contaminants are universally recognized. The findings from our study can provide guidance to countries and regions facing similar challenges, offering valuable information on treatment strategies and regulatory considerations. Additionally, understanding the regulatory landscape of emerging chemicals[64], from global agreements such as the Stockholm Convention to regional regulations like those in the United States, European Union, and China[65-67], is imperative. These disparities highlight the necessity for harmonization and enhanced coordination in regulatory frameworks to manage and mitigate the risks associated with emerging contaminants in drinking water effectively on a global scale. By acknowledging these geographic and regulatory complexities, our research contributes to broader dialogues on tackling emerging contaminants and advancing regulatory strategies to safeguard public health worldwide. This approach not only aids in the direct application of research findings but also fosters international cooperation and shared knowledge, essential for addressing global water quality challenges.

The research findings presented carry substantial implications for policy development, guidance on water treatment strategies, and decision-making processes regarding emerging chemical contaminants in drinking water. Firstly, the insights from our study can importantly inform the development and refinement of policies and regulations governing water treatment systems. By providing a comprehensive understanding of the broad impacts associated with various technologies, this research empowers policymakers to implement more targeted and effective regulatory measures that safeguard drinking water quality. Secondly, the research supports the selection and optimization of water treatment strategies. It demonstrates the effectiveness and sustainability of integrated approaches such as RBF-RO, especially in regions like the Netherlands, where surface water sources are prevalent. Furthermore, our developed WEALTH approach provides a systematic framework for decision-making, balancing the development of water purification systems, human health, and environmental impacts. This approach facilitates informed decision-making, ensuring that water treatment strategies not only meet societal

needs but also minimize adverse environmental consequences. Lastly, our study identifies key gaps and limitations that suggest potential areas for future research and investigation. Future studies could focus on exploring advanced treatment technologies, assessing the impacts of emerging chemicals on ecosystems, or enhancing the resilience and sustainability of water treatment systems in response to evolving water quality challenges. Overall, our research contributes to both scientific understanding and practical applications, benefiting society and the environment by providing actionable insights for managing emerging chemical contaminants in drinking water. These contributions are crucial for advancing regulatory strategies, optimizing treatment technologies, and ultimately protecting public health and ecological systems.

## Methods

### Simulation of the occurrence and removal of CECs

This study investigates the increasing impact of CECs on human health by examining their prevalence in drinking water treatment systems and assessing both their oncogenic and non-oncogenic effects. Supplementary Fig. 1 provides detailed illustrations of the water treatment processes, while Supplementary Table 1 outlines the critical process parameters. Given the emergent nature of CECs and the limited comprehensive studies on their toxicity and mechanisms of removal within water treatment systems, an extensive literature review was conducted to compile data on CEC concentrations in source water. In total, 93 CECs were analyzed, including 41 pesticides, 19 industrial chemicals, 17 pharmaceuticals, 7 antibiotics, 5 DBPs, and 4 PCPs, as detailed in Supplementary Table 2. Their concentrations in source water are summarized in Supplementary Table 9. To estimate residual concentrations after a series of treatments, initial concentrations were multiplied by cumulative removal rates, as documented in Supplementary Table 10 and presented in Supplementary Table 11. It is crucial to highlight that this modeling approach was rigorously evaluated by comparing it with an extensive body of experiment-based literature. The assessment outcomes are depicted in Supplementary Fig. 7, providing a comprehensive overview of the universality and robustness of our modeling approach in estimating real-world removal of CECs at the system level. This thorough assessment underscores the importance of integrating experimental data with model predictions to enhance the accuracy and reliability of findings related to the behavior and fate of CECs in water treatment processes.

### Modeling human health risks associated with drinking water consumption

In this study, the morbidity and mortality associated with specific water systems remain largely unquantified, even amidst extensive epidemiological research. To address this gap, the human health burden for each system was computed using simulated risk estimates, adhering to a quantitative environmental risk assessment framework[68]. The study employs an additive model to estimate risks associated with various CECs in drinking water. This model is chosen for its conservative nature, assuming that the combined effect of multiple contaminants is equal to the sum of their individual effects. The simplicity of the additive model is particularly practical when detailed interaction data among CECs are scarce, allowing for a more straightforward risk assessment based on available toxicity information. However, to ensure a robust analysis, the mixture toxicity estimates derived from the additive model were compared with outcomes calculated via dominant and multiplicative approaches, as depicted in Supplementary Fig. 8. Following these comparative analyses, the human health burdens of the assessed CECs due to drinking water consumption were quantified using DALYs, as advocated by the WHO. DALYs measure the total years of healthy life lost due to premature mortality and years lived with disability, facilitating a comprehensive evaluation that

encompasses both nonfatal and fatal health outcomes. This approach not only highlights the potential health impacts of contaminants in drinking water but also provides a basis for prioritizing interventions and public health measures to mitigate these risks effectively. In this study, DALYs were computed using a spreadsheet based USEtox model, founded on scientific consensus for characterizing the human and ecotoxicological impacts of chemicals[69]. While the USEtox model commonly factors in intake through inhalation and ingestion in intake fraction ($IF$) calculations, this case study solely focused on the calculation of direct ingestion of drinking water ($IF^{ing}$). This decision stemmed from the recognition that the health risks associated with inhalation exposure to waterborne CECs in drinking water are comparatively minor when juxtaposed with those of the ingestion route. Thus, the human health effect ($HE$) of each CEC was estimated using Eq. (1):

$$HE_i^j = \sum_i \left( EF_i^j \times DF^j \times IF_i^{ing} \right) \qquad (1)$$

Where subscript $i$ represents a specific CEC in the source water ($i = 1, 2, 3,..., I$; with $I = 93$ in this study, as detailed in Supplementary Table 2), superscript $j$ represents the type of $HE$ ($j$ denotes non-cancer or cancer effects), $EF_i^j$ denotes the mid-point human toxicity factor reflecting the change in the lifetime probability of disease $j$ due to the variation in the lifetime intake of chemical $i$ (expressed in cases per kilogram; see Supplementary Table 4), $DF^j$ represents the mid-point human toxicity factor concerning disease cases, incorporating years of life lost and years of life disabled (with $DF$ being equal to 2.7 DALYs per case and 11.5 DALYs per case for non-cancer and cancer effects, respectively), and $IF_i^{ing}$ (as per Eq. (2)) denotes the quantity of ingested chemical $i$ (expressed in kilograms) from consuming drinking water from a drinking water production plant with a 25-year service life.

$$IF_i^{ing} = SP \times IR_{ing} \times CE_i \qquad (2)$$

Where $SP$ represents the total population served by the drinking water production plant (in this study, $SP = 500,000$), $IR_{ing}$ denotes the cumulative volume of drinking water consumed per capita over a 25-year exposure period to waterborne chemicals, assuming an average daily ingestion rate of 1.4 liters[69], and $CE_i$ (as per Eq. (3)) is the exposure concentration of chemical $i$. In the case study, $CE_i$ refers to the residual concentration of chemical i in the source water or water after a series of treatments.

$$CE_i = CS_i \times \prod_k (1 - R_i^k) \qquad (3)$$

Where $CS_i$ indicates the initial concentration of chemical $i$ in the source water, as reported in the literature. When concentrations of CECs fell below the limit of detection (LOD) or limit of quantification (LOQ) in the original studies, a conservative approach was adopted. Specifically, these values were assigned as equivalent to half of the LOD or LOQ, as detailed in Supplementary Table 9. $R_i^k$ represents the efficiency of water production unit $k$ in removing chemical $i$ (with $k = 1, 2, 3, ..., K$; in this study, K was equal to 6 and 7 for the RBF-RO and RBF-ET systems, respectively, as specified in Supplementary Table 10). In cases where data on the removal efficiency of a particular chemical in a treatment process unit were lacking, the average removal efficiency for the same type of chemical (e.g., pesticides, industrial chemicals, pharmaceuticals, antibiotics, DBPs, or PCPs in this study) in the same process unit was utilized.

### Life-cycle inventory data acquisition and impact assessment

The life-cycle environmental effects of each water production system in this study were evaluated following the general steps outlined in ISO

14040. For this case study, the functional unit was defined as the production of 1 cubic meter (m³) of drinking water over a 25-year operational period, reflecting the typical lifespan of engineered water treatment infrastructure[70]. The analysis focused on the operational phase, excluding facility construction and decommissioning due to their minimal impact compared to long-term operations[71]. Comparisons between the two water production systems incorporated both foreground processes (such as water production, RO-concentrate treatment, and sludge disposal) and background processes (including the use of energy, chemicals, and other materials in both on-site and off-site foreground processes). Transport-related impacts were excluded, based on the assumption of identical transport distances for all systems and an estimated population of approximately half a million inhabitants per service area, derived from a per capita daily water usage of 120 l[26]. Foreground inventory data for the alternative systems, which included operational energy, chemicals, consumables, waste streams, and gaseous emissions, primarily sourced from literature, are detailed in Supplementary Table 12. Whereas Supplementary Table 13 provides inventory data concerning the transportation of chemicals and consumables from manufacturers to the water production plant. Background inventory data on chemicals, energy, and materials, sourced from the Ecoinvent database, were selected to reflect the local electricity mix and specific material and energy inputs relevant to the case study location, as outlined in Supplementary Table 14.

In this study, life-cycle inventory inputs and emissions were transformed into 10 environmental impact categories, utilizing characterization factors from the Hierarchist ReCiPe 2016 midpoint-based methods version 1.01, accessed via SimaPro software. The impact categories selected for analysis included global warming, ozone depletion, terrestrial acidification, freshwater eutrophication, marine eutrophication, terrestrial ecotoxicity, freshwater ecotoxicity, marine ecotoxicity, mineral consumption, and fossil fuel depletion, as shown in Supplementary Table 15. These categories represent commonly included environmental impacts in LCAs for urban water management[33,72]. Importantly, the life-cycle inventory data for all process configurations within each system served as model inputs for subsequent impact calculations. Therefore, the model outputs presented here pertain to the overall results of each system rather than specific process configurations. This comprehensive approach allows for a detailed comparison and evaluation of the environmental impacts associated with different water production systems, providing valuable insights into their sustainability and efficiency.

## Assessment of RBF-RO in other countries

To assess the potential impact of site-specific electricity mixes on the environmental performance of the RBF-RO system, we analyzed the energy structures across 136 countries and territories, hereafter referred to as "countries," covering regions such as Africa, the Americas, Oceania, and Europe. These countries were selected to represent a great portion of the global population, accounting for over 90% of the world's inhabitants, thus ensuring the representativeness and global relevance of our analysis. The life-cycle inventory associated with electricity generation for each country was derived from predefined unit processes within the Ecoinvent database. This integration into the LCA model facilitated a comprehensive analysis and comparison of how different electricity generation mixes affect the environmental outcomes of the RBF-RO water treatment system. The specific Ecoinvent processes utilized for each country's electricity generation are detailed in Supplementary Table 16. This approach allowed us to understand the variance in environmental impacts driven by different energy sources and efficiencies used across diverse geographical contexts. By considering the specific electricity mixes of each country, the study not only provides insights into the

environmental impacts associated with the operational phase of the RBF-RO system but also highlights the importance of considering regional energy policies and practices in environmental performance assessments. This nuanced understanding is crucial for policymakers, environmental scientists, and engineers aiming to optimize water treatment solutions in a way that aligns with both local energy landscapes and global sustainability goals.

## Variability, uncertainty and sensitivity analysis

A MC simulation was employed to quantify input uncertainties and to explore the potential variability and uncertainty in the health disease burden estimates associated with each water treatment system. The simulation assumed the mutual independence of water treatment methodologies, with the necessary integrated assumptions, except for the variability inherent in the Ecoinvent database. Each input parameter was assigned a probability distribution, with maximum and minimum plausible values derived from this study, as outlined in Supplementary Tables 9 and 10. Due to data scarcity—a common challenge in such research—the true underlying distribution of several parameters could not be fully characterized[26]. Therefore, triangular distributions were utilized, using the mean and extreme values of available data to establish the mode and the minimum or maximum values, respectively. The central tendency results were compared with the variability at the 5th and 95th percentiles, derived from the distribution of outcomes following 10,000 MC iterations, as dictated by Eq. (1). Mean values, along with their 95% confidential intervals, were utilized to reflect the inherent uncertainties in the estimates. Acknowledging the significance of uncertainty in the quantification and qualification of health disease burdens and life cycle impact assessments, the absence of specific input parameter ranges in broader evaluations can complicate the precise quantification and mitigation of uncertainties, particularly when compared to simulation-based methodologies. Our strategic emphasis on water treatment process simulations reflects a deliberate effort to enhance knowledge and practices within this critical aspect of environmental engineering. To ensure the robustness of our findings, the simulation was repeated to verify the adequacy of the number of simulations for achieving reproducible results. The uncertainty analysis was conducted using Oracle Crystal Ball.

To identify the pivotal factors influencing trends in health and environmental effects attributable to water purification practices, a sensitivity analysis was performed on key model parameters. These included the influent concentration of CECs, the removal efficiencies of system components, and the inputs and emissions throughout the treatment processes. Initially, based on the MC-generated health disease burden simulation sets, Spearman's rank-order correlation was used to execute regression correlation analysis between outcomes for cancer and non-cancer disease burdens and the input parameters, which encompassed influent CEC concentrations in source water and RBF removal rates. For environmental impact assessments, each model input of the RBF-RO system was systematically varied by ±20% (as referenced in Supplementary Tables 12 and 13), with subsequent observation of the resultant impact on the environmental impact categories through a one-factor-at-a-time sensitivity analysis. A sensitivity coefficient was formulated, relating the ratio of the change in the output parameter to the change in the input parameter[26]. The findings from the sensitivity analysis are presented in Supplementary Figs. 5 and 6, offering a comprehensive view of the factors most critical to the environmental and health outcomes of the water purification processes.

## Data availability

All model inputs used for the analysis in this study are publicly available through the cited literature or the data provided in the Supplementary Data 1 and Supplementary Information.

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

## Acknowledgements

The authors acknowledge funding support from the National Natural Science Foundation of China (51922013; 52321005), Shenzhen Key Research Project (GXWD20220818172959001; KCXST20221021111404011), Shenzhen High-level Talent Team Project (KQTD201909209172630447), and Fundamental Research Funds for the Central Universities (Grant No. HIT.OCEF.2023049). The authors thank Dr. Walter G.J. van der Meer from University of Twente and Dr. Gang Liu from Research Center for Eco-Environmental Sciences, the Chinese Academy of Sciences for providing preliminary information regarding water production processes in the Netherlands.

## Author contributions

X.W. conceived and led the study. Y.L.L. and Y.R.P. contributed equally to the model formulation and system analysis. X.W., Y.L.L., Y.R.P., Z.Y.W., J.X.M., and L.H.T. performed the data analysis and visualization. Y.L.L. and Y.R.P. wrote the paper. X.W., G.D., J.L, N.-Q.R., and D.B. contributed substantially by commenting on and revising the text.

## Competing interests

The authors declare no competing interests.
