## [Transparent Peer Review file · Nature Communications]

Leveraging the water-environment-health nexus to characterize sustainable water purification solutions

Corresponding Author: Professor Xu Wang

Version 0:

Reviewer comments:

Reviewer #2

(Remarks to the Author)

General: This paper evaluated the removal efficiency of 93 contaminants of emerging concern (CECs), associated disease burden, and resulting environmental impacts of two water treatment alternatives, riverbank filtration (RBF)-extended treatment (ET) train and RBF-reverse osmosis (RO) considering different energy mix in 136 countries. The work is original in terms of connecting water treatment, human health, and environmental impacts together to evaluate water treatment alternatives. In terms of methodology, enough detail is provided for the work to be reproduced. The process modeling for estimating CECs removal, however, is overly simplified using the cumulative removal rates. This approach assumes that unit processes in the treatment train are independent in terms of CEC removal. Each unit process, however, will change multiple water quality parameters and influence the removal efficiency of the subsequent process. For example, the removal efficiency of a CEC through granular activated carbon depends on existing co-contaminants in the influent water that is affected by the previous process. Since the study evaluated only two treatment trains, the actual process models should be developed to simulate the water quality change through the system. The results are limited to the two alternatives evaluated, and the generalization of the finding related to water treatment processes, such as “synergy between ecological processes and high-throughput membrane technology could produce healthily and environmentally favorable outcomes than status quo paradigms based on technologies in series”, is questionable. If this is a hypothesis to be tested, the alternative water treatment trains should be carefully designed to reflect different combinations, for example, a treatment train including non-RBF processes and high-throughput membrane technology. The evaluation of such alternatives will help determine “the extent to which synergies can be built between natural ecological processes and alternative engineered trains” as claimed in Introduction. Some specific comments are provided below.

Specific comments

Lines 20-23: The study didn't consider different water purification alternatives in 136 countries, but different energy mix in 136 countries. This statement is misleading. In reality, this study evaluated two water treatment alternatives, RBF-ET and RBF-RO. Similarly, the study didn't consider different ecological processes, but only one ecological process, namely riverbank filtration. The abstract should be revised to accurately reflect the scope of the work.

Lines 37-40: A brief summary of commonly employed treatment trains for water treatment should be provided to support the design of water treatment alternatives in this study.

Line 40: What are criteria to be considered sustainable water treatment?

Lines 65-66: The current study is not well designed to quantify “the extent to which synergies can be built between natural ecological processes and alternative engineered trains”.

Lines 77-83: The results are specific to this treatment train and how can that be generalized? What are typical ecological-based treatment processes? What's the removal range of different contaminants in those processes? What are typical non-RBF treatment trains for drinking water? Supplementary Fig. 1 caption mislabeled a and b.

Lines 167-169: How the transportation distance is determined?

Line 190: How much reduction?

Lines 222-226: This statement is overly generalized from the study results since this study evaluated only two specific treatment trains with both incorporating RBF.

Lines 245-248: Fig. 6 does not provide much information. The potential trade-offs, co-benefits and optimization strategies should be highlighted in the figure.

Lines 256-257: How synergy is quantified?

Lines 288-291: Such uncertainties should be evaluated in the study since it will highly influence the cumulative removal rate.

Lines 304-307: See the general comment in terms of determining CECs removal.

Lines 390-392: Uncertainties occurs not only in treatment performance, but also the risk quantification especially cumulative risks, and life cycle assessment. How the uncertainties are considered in those assessments?

Reviewer #3

(Remarks to the Author)

In this paper titled "Leveraging the water-environment-health nexus to characterize sustainable water purification solutions" the authors have modelled the human and environmental health implications of removing chemicals of emerging concern by an extended treatment (ET) train and advanced drinking water treatment by reverse osmosis (RO) using a natural riverbank filtrate (RBF) as source water. The authors also performed an LCA. I find that the manuscript is well written and presents interesting results. However, the novelty aspect is currently unclear, and a true novelty statement is lacking. The literature is populated with many studies that tackled the health implication and LCA of drinking water treatments, so the author should clearly state how this work is novel compared to the literature. Furthermore, I have some major concerns about the technical aspect. First, the authors compared ET and RO, but the operating conditions and RO configuration are not immediately available to the reader. In the case of RO, several solute and membrane parameters influence the extent of removal of organic chemicals (see DOI 10.1016/j.watres.2004.03.034 for an old, yet solid literature review). Therefore, the results presently included in the manuscript may or may not be applicable to other RO drinking water treatment plants. This aspect and its implications should be thoroughly discussed. Second, the authors evaluated the environmental impact of RBF-RO in 136 countries considering site-specific electricity generation mixes. In the manuscript introduction, the authors mention differences between bank filtration in humid and arid regions. Based on these differences, it is unclear whether RBF-RO may be the best approach in all countries that have been considered for the LCA. Additionally, the amount of energy used by RO also depends on the quality of the feed water (i.e., higher solute content -> higher operating pressure -> higher energy consumption). This aspect should be included in the model or at least acknowledged and discussed. Third, the environmental impact of treating RO waste stream (i.e., RO concentrate or brine) is mentioned but not clearly presented or discussed. Production of brines is one of the major drawbacks of RO treatment and I think it should be more prominent in the paper. Finally, I feel that additional discussions are needed to address the relevance of the set of chemicals considered for disease burden modeling for other sites worldwide. The composition of the mixtures of pollutants that occur in drinking water sources can greatly vary depending on site-specific chemical use, chemical persistence and mobility, and regulatory actions (e.g., requirements for wastewater treatment or compound-specific bans). I am also surprised to see that PFAS were not considered, as the proximity of chemical manufacturing sites to drinking water sources has been an issue in the Netherlands for decades (see <https://pubs.acs.org/doi/10.1021/acs.est.2c06015>). Please find my specific comments below:

Line 28: rather than "taking advantage", I would say that modern society has "relied" on rapid developments in chemistry, trusting the promise of better living through chemistry.

Line 29: I feel it would be valuable to add data on the amount of chemicals produced each year, for example a mass range.

Line 30-32: please include additional references

Line 36: please define a representative range for low concentrations and clarify somewhere in this paragraph that the CECs you are referring to are organic compounds.

Line 43: please provide an example of aggravated contamination issues resulting from chemical-intensive water treatment

Line 45: you claim that "most" water treatment technologies are inspired by nature, but you provided only one reference relevant for RBF. What about other technologies? For example, are RO, GAC, and UV disinfection also inspired by nature?

Line 49-50: please mention which processes are responsible for attenuation of pollutants, where they exactly occur in a riverbank filtration system, and which compound physicochemical properties may lead to complete/incomplete removal. Infiltration per se does not necessarily imply attenuation.

Line 51: If a trace organic chemical is not regulated, is further drinking water treatment still required?

Line 53: please define the term "micropollutants"

Line 54: please clarify what "technical viability" means

Line 56: Is molecular diameter the only chemical descriptor relevant to removal by RO membranes?

Line 59: please provide a molecular weight range.

Line 92: how are system efficiency and chemical removal efficiency different from each other? Please clarify.

Line 93: please clarify whether the CECs selected for this study are representative for a broader set of CECs (in terms of physicochemical properties and source water concentrations). Additionally, are these chemicals from a Dutch drinking water source representative for pollution only in the Netherlands? Or the whole Europe? Or developed Western countries? It would be great if you could mention whether these chemicals are regulated or not (either locally or at EU level).

Line 122: "single RBF system" means natural treatment solely relying on bank filtration, correct?

Line 125: when using acronyms (e.g., ppy), please clarify what they stand for and stick to them for the rest of the manuscript.

Line 126: somewhere in the discussion, could you please elaborate more on why (carcinogenic) DBPs represented the main category with decreasing trend?

Line 149: Was the impact of RO concentrate considered at all?

Figure 1: please explain the acronyms in the figure captions

Line 208: please provide an example of clean coal technology

Line 210: please clarify what's lignite and its role

Line 218-219: consider mentioning that this need for research is currently being tackled by the exposomics community

Line 217: are these figures in line with other research or is there a range we can provide?

Line 218: I'd say that there is a need for more research as studies on this topic are being conducted. So, either acknowledge this fact or include (more) references

Line 219: Why do they occur at extremely low concentrations? While not easy, I disagree with the statement about CECs being difficult to detect, as a variety of methods (both for sample prep and analysis) are readily available. Please consider revising this sentence.

Line 223: is "ecological processes" the most appropriate term here? Are you referring to bank filtration?

Line 228: This finding is not really novel, so I'd clarify how this aligns with the scientific literature.

Line 251: Again, this is not really a novel finding as the chemical removal efficiency of advance treatment technologies is widely documented. Also, define "innovative technologies".

Line 279: I find that the term "ignorant" bears quite some negativity with it and perhaps it would be best to say that emerging threats are often neglected.

Line 300: While true, a variety of sources for organic chemical toxicity are available. Have you considered looking into bioactivity metadata in PubChem or the EPA CompTox Chemicals Dashboard?

Line 344-347: For how many chemicals were removal data not available? Also, I am not sure about the approach you used here, chemicals in the same category can greatly differ from each other in terms of physicochemical properties and consequently degree of removal. I think a chemical structure-based approach may have been more appropriate. Just as an example looking at the chemicals in your dataset, 1,4-dichlorobenzene and dimethoate are both pesticides, but are not related except for the category they were assigned to. The former is aromatic, non-polar, and has a molecular weight lower than most commercial RO membranes molecular weight cut-off. Conversely, the latter is a highly polar organophosphate, non-aromatic, and has a molecular weight higher than the membranes' cut-off. Therefore, great differences in removal rates can be expected for these two compounds, meaning they'll contribute to substantial shifts in the mean removal value in that particular category.

Supplementary Information

Figure 1b caption: stating that RO removes "a lot of substances/waterborne contaminants" is not too scientifically sound, please improve where needed. Furthermore, I find it contradictory to mention that that RO "leaves pure water only" and then explain in the next sentence that additional treatments are needed to remove residual aluminum, carbon dioxide and methane. Furthermore, I believe it's important to provide more details about the drinking water treatment parameters and operating conditions. For example, in the case of RO, chemical removal efficiency can greatly vary depending on several factors including but not limited to membrane polymer type (polyamide, cellulose acetate), membrane configuration (spiral wound, hollow fiber), membrane molecular weight cut-off, number of membranes, RO recovery, operating pressure, etc.

Supplementary Table 1: In addition to CAS numbers, please include machine-readable structure identifiers for each chemical to enable further modeling. Either SMILES or InChIKey work great.

Supplementary Table 2: please clarify why toxicity data were not applicable in some case.

Reviewer #4

(Remarks to the Author)

- 1) The title "Leveraging the water-environment-health nexus to characterize sustainable water purification solutions" is in the line of the conducted research.
- 2) Abstract is too long. This may be improved by focusing on novel methodology used, key results and future perspective of the finding.
- 3) The objectives of the project may clearly be mentioned and the findings may be aligned in the same sequence.
- 4) Considering line # 77. The "Overview of the water purification systems" does not seem the part of the findings. It is better to explain this in an appropriate place.
- 5) The discourse of the manuscript needs improvement.
- 6) Methodology used must be clear enough to understand by the reader,.
- 7) The "References" may be updated.
- 8) CONCLUSION:
The research is applied and useful thus may be accepted for publication after improvement as suggested.

Prof Dr Muhammad Fahem Malik
Dated: May 18. 2023

Reviewer #5

(Remarks to the Author)
GENERAL COMMENTS

The paper deals with a comparison between different in series purification systems (RBF, RBF-ET and RBF-RO) for providing drinkable water, to assess the most sustainable in terms of both human burden disease, toxicity for terrestrial, freshwater, and marine ecosystems, and broad environmental impacts (global warming, stratospheric ozone depletion, terrestrial acidification, mineral resource consumption, fossil fuel depletion). The Authors conduct a simulation by Monte Carlo by using data of occurrence of 93 CECs from literature and toxicological data for carcinogenic and not carcinogenic effects.

The key message is the proposal for an integrated approach that strikes a balance between the development of advanced technology of the water purification system, human health, and the broad impacts on the environment, what the Authors name the WEALTH approach

Noteworthy results are that the RBF-RO system, compared to the RBF or RBF-ET, reduce the negative impact of the occurrence in water of 93 CECs on human health and environment. Noteworthy results, are also those showing how different contexts in terms of energy mix can affect the broad environmental impacts due the different water treatment systems.

The work is significant because it deals with an important and still little-known topic, which is the health impacts of emerging contaminants on human health and the environment. In addition, the work is significant because processes a large amount of data and consider site-specific data on 136 countries, suggesting solutions in specific context. It expands the results of a previous published study of the same group on the human health and environmental impacts of different solutions of water resource recovery. The work supports the conclusions and claims.

A flaw in the data analysis is the lack of an assessment or discussion of the uncertainty of the models estimating the global burden of DALYs. As consequence the significance of the variations between the DALYs for the different scenarios is not assessable. This aspect would require revision.

The methodology is sound because the Authors use internationally standardised methodologies, however, the limitations and uncertainties of these methodologies should be mentioned and discussed in the paper.

Overall there is enough detail in the methods provided to reproduce the work even though some input variables are not made explicit, and more details for the calculation of the DALYs should be provided.

SPECIFIC COMMENTS

Results

Overview of the water purification systems.

This part doesn't really seem appropriate for the results section. I would suggest splitting it between the introductory part and/or material and methods.

Modeled disease burdens associated with CECs in each water production system

- The Authors calculate the burden of disease using DALYs, which should be expressed in "years", but then they compare this amount with the tolerable health risk of 1.00×10^{-6} person⁻¹ year⁻¹. The link between these entities should be better clarified. In addition reference 22 seems to be not appropriate.
- There is no assessment of the uncertainty of the estimates of the disease burden.
- Supplemental Table 3: How the toxicity factors were derived/calculated?

Environmental impacts throughout the water treatment process

The Authors use the life-cycle assessment (LCA) to assess the environmental impact of the different water purification systems. However, although LCA is an internationally standardised methodology, the article could be improved if some additional information were given, e.g. how the ecotoxicity is modelled? What is the meaning of 1,4-DCB units?

Discussion

Line 218: references 30 and 31 do not seem to be appropriate (I think there was a shift in the reference list)

Methods

Simulation of the occurrence and removal of CECs.

Supplemental Table 8: concentration values of ECCs in water were derived from literature. There were, in the original papers, concentrations below the limit of detection (LOD) or quantification (LOQ)? How did the authors handle concentrations below LOD/LOQ?

Modeling human health risks associated with drinking water consumption

The Authors do not take into consideration inhalation as exposure pathway, however inhalation can be significant for some very volatile contaminants.

Line 336: What is the value used for ingested drinking water used in equation 2.

Reviewer #6

(Remarks to the Author)

I have thoroughly reviewed your paper and found it to be of significant importance to the field. The study addresses an important topic concerning the occurrence and health impacts of emerging contaminants in drinking water, as well as the environmental performance of water production systems. I appreciate the comprehensive review of the existing literature and the rigorous assessment of health impacts using DALY burden of diseases.

Overall, the manuscript is well-written and organized. The introduction provides a clear background and rationale for the study, highlighting the importance of addressing emerging contaminants and their potential risks to human health and the environment. The methods section is detailed and comprehensive, allowing for reproducibility of the study. The use of the USEtox model for estimating health impacts and the life cycle assessment for evaluating environmental performance are appropriate and add credibility to the findings.

The results section presents the key findings in a clear and concise manner, supported by diverse figures and tables. The discussion section provides a thorough analysis and interpretation of the results, highlighting the implications and significance of the study. The conclusions are well-supported by the data and align with the objectives of the study.

The study provides valuable insights into the assessment of emerging contaminants in drinking water and offers a comparative analysis of different water production systems. It contributes to the existing literature in a meaningful way and will be of interest to researchers and practitioners in the field. I believe this manuscript is suitable for publication in our journal.

However, there are a few minor revisions that I recommend addressing before publication:

1. In the methods section, it would be helpful to provide additional information on the selection criteria for the included contaminants. Clarify if any specific criteria were used to prioritize the contaminants included in the analysis.
2. Provide more details on the uncertainty analysis conducted in the study. Describe the methodology used for the Monte Carlo analysis and how the uncertainty ranges were determined.
3. Consider discussing the limitations of the study in more detail. While the manuscript briefly mentions some limitations, a more comprehensive discussion in a separate paragraph would enhance the robustness of the study.
4. The implications and practical applications of the findings could be further highlighted in the discussion section. Discuss how the results of this study can inform decision-making processes, water treatment strategies, and policy development.

Version 1:

Reviewer comments:

Reviewer #5

(Remarks to the Author)

The Authors answered to all my comments thoroughly and revised the paper. They added the paragraph: "Sensitivity analysis of disease burdens and environmental impacts" and extended considerably the discussion, however not always improving clarity.

There remains an issue where the Authors' answer is not completely comprehensive, and it is the assessment of the uncertainty of the estimates. In fact, although the authors were comprehensive in stating all the elements of uncertainty in the estimates of DALYs, my comment referred to a quantitative assessment of the uncertainty in terms of the confidence intervals of the estimates. The Authors stated they used the Monte Carlo simulation to account for uncertainty but from my understanding by this technique they just incorporated variability.

Reviewer #7

(Remarks to the Author)

I have thoroughly reviewed the manuscript and supplementary files as well as the authors' response to reviewers letter. This is a detailed and highly relevant paper that contributes to the current literature in a meaningful way. This study examines the human-environment-water nexus by analyzing the presence, occurrence, and subsequent health impacts of contaminants of emerging concern in various water sources and determining the efficiency of different purification systems for drinking water. The manuscript is well-written, concise, and the authors appear to have taken a considerable amount of effort and time to address each of the reviewers' extensive comments in detail, while still remaining within scope of the originally proposed work. Methods are sound and presented in a manner that supports reproducibility. This study provides a foundation for future work to investigate more nuanced occurrences with regard to water quality parameters and impacts from CECs in various matrices, as noted in other comments by the other reviewers.

I recommend this manuscript is fit for publication at this stage, with no further edits or comments to provide that would truly add to the presented work.

Reviewer #8

(Remarks to the Author)

While the authors have addressed the comments of the reviewers especially in terms of limitations of the study/modelling approach. However, the authors have not addressed the limitations of the study in the conclusion or abstract. It is imperative that conclusion clearly list out the limitations in making such conclusions. For example, better to soften the conclusion by incorporating phrases like "by analyzing two treatments trains". while I believe riverbank filtration is a strong candidate, discounting others or adding RO to this can be problematic, especially there can be instances where RBF or RBF-RO are not suitable. with the limitations of the study approach regarding the influences of the matrix of the water as well as influence of one process on subsequent processes and variability that can occur in various countries, the conclusion should still be softened. Your approach is novel but that does not mean you can conclude that it is the only solution.

Version 2:

Reviewer comments:

Reviewer #5

(Remarks to the Author)

The authors answered clarifying how their approach captured both variability and uncertainty of their final estimates. They answered comprehensively to my comment.

Response to Reviewer 2

1. This paper evaluated the removal efficiency of 93 chemicals of emerging concern (CECs), associated disease burden, and resulting environmental impacts of two water treatment alternatives, riverbank filtration (RBF)-extended treatment (ET) train and RBF-reverse osmosis (RO) considering different energy mix in 136 countries. The work is original in terms of connecting water treatment, human health, and environmental impacts together to evaluate water treatment alternatives. In terms of methodology, enough detail is provided for the work to be reproduced.

R: We are grateful for the reviewer's commendation regarding the originality of our work. Additionally, we value the insightful comments and suggestions, which have not only enabled us to enhance our manuscript but have also sparked intriguing ideas for future research endeavors. We have made substantial revisions to the manuscript based on this feedback. Below, we provide a detailed point-by-point response to each of the comments raised.

2. The process modeling for estimating CECs removal, however, is overly simplified using the cumulative removal rates. This approach assumes that unit processes in the treatment train are independent in terms of CEC removal. Each unit process, however, will change multiple water quality parameters and influence the removal efficiency of the subsequent process. For example, the removal efficiency of a CEC through granular activated carbon depends on existing co-contaminants in the influent water that is affected by the previous process.

R: We appreciate the reviewer's concern regarding the modeling approach employed in our study to estimate the removal of CECs) across various process configurations. The reviewer astutely points out the necessity of considering potential interdependencies among unit processes within the treatment train and their impacts on CEC removal. In response, we have rigorously verified our modeling approach through comparison with a substantial body of experimental literature. These verification results are included in the Supplementary Information, providing a comprehensive overview of the universality and robustness of our modeling approach in estimating real-world CEC removal at the system level (refer to Supplementary Information, Page 8, Supplementary Fig. 7). Furthermore, the rationale for our modeling approach is detailed in the revised manuscript (Main text, Pages 21–22, Lines 515–520).

Nevertheless, we have taken the reviewer's critique seriously and acknowledge that our initial modeling might not fully capture the complex interplay of quality

parameters and treatment processes. Furthermore, we recognize that variations in water quality parameters induced by one unit process can influence subsequent processes, thereby underscoring the significance of considering system complexities in our modeling approach. Moreover, we acknowledge that the removal mechanisms of different CECs in various process configurations represent an ongoing area of research and remain a significant knowledge gap. While our study is designed to provide a valuable initial framework for estimating CEC removal at the system level, we agree that further research is needed to develop more sophisticated models that explicitly account for the intricate interactions and interdependencies between unit processes. To reflect these insights, we have added detailed discussions to the revised manuscript (Main text, Page 18, Lines 420–436), addressing the need for more complex modeling approaches and the importance of further research in this area.

3. Since the study evaluated only two treatment trains, the actual process models should be developed to simulate the water quality change through the system. The results are limited to the two alternatives evaluated, and the generalization of the finding related to water treatment processes, such as “synergy between ecological processes and high-throughput membrane technology could produce healthily and environmentally favorable outcomes than status quo paradigms based on technologies in series”, is questionable. If this is a hypothesis to be tested, the alternative water treatment trains should be carefully designed to reflect different combinations, for example, a treatment train including non-RBF processes and high-throughput membrane technology. The evaluation of such alternatives will help determine “the extent to which synergies can be built between natural ecological processes and alternative engineered trains” as claimed in Introduction.

R: We are grateful for the reviewer's insightful feedback. Despite the established reliability of our modeling approach, we wish to offer further justification for our decision not to develop actual process models in response to the suggestion. Currently, significant constraints hinder our ability to conduct additional experiments and gather new data necessary for developing such models. Resource limitations, particularly in terms of time and budget, have necessitated prioritizing the optimization of existing literature and data-driven modeling approaches to achieve our research objectives. Our reliance on a combination of datasets, assumptions, and calculations, rather than on developing actual process models, aligns with established practices in the field of water treatment research. Numerous studies utilize data-driven modeling techniques to assess water quality changes and treatment outcomes at the system level, especially when conducting larger-scale experiments is challenging or resource-intensive. The

universality and robustness of our modeling approach have been thoroughly assessed, validating the estimation of CEC removal efficiencies. This validation process ensures the reliability and relevance of our results, even without the development of actual process models. Given these considerations, we believe our developed modeling approach, with its demonstrated universality and robustness, provides a solid foundation for this study. While developing actual process models could add depth to our research, it is not essential for achieving our stated objectives and may not be feasible under the current constraints. Nevertheless, we have taken the reviewer's suggestion seriously and have included additional analysis to justify the reliability of our modeling approach (see Supplementary Information, Page 8, Supplementary Fig. 7) and to acknowledge the potential merits of actual process models in future studies (Main text, Page 18, Lines 423–436).

Subsequently, we address the reviewer's comments regarding the generalization of our findings and the clarity of our expression. We acknowledge the concern that our results may be limited to the two treatment alternatives evaluated, and that our initial expression, particularly the claim of a 'synergy between ecological processes and high-throughput membrane technology', may have been unclear or overly generalized. In response to these concerns, we have made significant revisions to our manuscript. The revised manuscript now features a more precise and nuanced expression of our findings (Main text, Page 2, Lines 21–22; Page 4, Lines 70–72; Page 13, Lines 299–301). This revision ensures that our claims are accurately conveyed and understood, providing a clearer and more detailed explanation of our research outcomes.

Moreover, while our study was focused on evaluating two specific treatment alternatives, we have revised the relevant sections of our manuscript to clarify that our findings are specific to the conditions and parameters tested. We emphasize the need for caution in generalizing our results to different treatment scenarios. Specifically, we have expanded the discussion on the potential variability of outcomes in alternative water treatment configurations, highlighting the importance of careful design and consideration of specific operational conditions (Main text, Page 14, Lines 331–336). Although additional experiments and simulations are not feasible at this time, we have underscored the importance of future research that explores a broader range of treatment alternatives. We encourage the scientific community to conduct such studies to further investigate the extent to which synergies can be developed between natural ecological processes and alternative engineered systems (Main text, Page 12, Lines 322–331). We are confident that these revisions will enhance the clarity and specificity of our findings, effectively addressing your concerns and providing a more

accurate representation of our research.

4. Lines 20-23: The study didn't consider different water purification alternatives in 136 countries, but different energy mix in 136 countries. This statement is misleading. In reality, this study evaluated two water treatment alternatives, RBF-ET and RBF-RO. Similarly, the study didn't consider different ecological processes, but only one ecological process, namely riverbank filtration. The abstract should be revised to accurately reflect the scope of the work.

R: We appreciate the reviewer's concern regarding the representation of our study's scope in the abstract. We understand that the original wording may have inadvertently led to confusion, and we sincerely apologize for any misunderstanding this caused. The reviewer correctly notes that our study evaluated two specific water treatment alternatives operated under various energy mixes in 136 countries. It is important to clarify that this was not an exhaustive analysis of all available purification methods across these nations. We value this observation and have revised the abstract accordingly to prevent any overgeneralization. Furthermore, we acknowledge the reviewer's insightful observation that our study predominantly focused on riverbank filtration as the ecological process under investigation. This specific focus was indeed central to our research and should have been more explicitly communicated in the abstract. In response to these insights, we have made careful revisions to the abstract to reflect the true scope and focus of our work more accurately. The revised abstract now clearly delineates the specifics of the treatment alternatives we assessed and emphasizes our focused examination of riverbank filtration rather than other ecological processes (Main text, Page 2, Lines 16–28).

5. Lines 37-40: A brief summary of commonly employed treatment trains for water treatment should be provided to support the design of water treatment alternatives in this study.

R: We appreciate the reviewer's suggestion and have acted accordingly. The relevant statements in the manuscript have been carefully rewritten to reflect this guidance (Main text, Page 3, Lines 43–47).

6. Line 40: What are criteria to be considered sustainable water treatment?

R: We appreciate the reviewer's question. In response, we have reconstructed the relevant sentence to enhance clarity (Main text, Page 3, Lines 49–51).

7. Lines 65-66: The current study is not well designed to quantify “the extent to which

synergies can be built between natural ecological processes and alternative engineered trains”.

R: We value the reviewer's question and have accordingly reconstructed the relevant sentence to improve clarity (Main text, Page 3, Lines 71–72).

8. Lines 77-83: The results are specific to this treatment train and how can that be generalized? What are typical ecological-based treatment processes? What's the removal range of different contaminants in those processes? What are typical non-RBF treatment trains for drinking water? Supplementary Fig. 1 caption mislabeled ‘a’ and ‘b’.

R: We acknowledge the reviewer's concerns regarding the specificity of our results to the treatment trains presented in the manuscript. We would like to clarify that these treatment trains are utilized as case studies to illustrate the potential of integrating ecological-based and technological alternatives in mitigating adverse effects within the water-health-environment nexus. This approach enhances our understanding and improvement of the co-benefits among clean water supply, public health protection, and environmental sustainability goals. Specifically, our study modeled the health effects and broad environmental impacts of reducing a variety of CECs from drinking water through the combined use of riverbank filtration and engineered measures. While we appreciate the reviewer's interest in the removal ranges of different contaminants through other ecological-based treatment processes for drinking water, an extensive investigation into these processes is beyond the scope of our current work. However, to ensure clarity regarding the scope and goals of our study, we have revised the relevant sections of our manuscript (Main text, Pages 4–5, Lines 70–82). Additionally, we acknowledge an oversight in the labeling of the caption in Supplementary Fig. 1, where labels 'a' and 'b' were inadvertently mislabeled. We sincerely apologize for this error and have corrected the mislabeled captions in the Supplementary Materials. We have also conducted a thorough review of all figures and their captions to ensure accuracy in the resubmission (Supplementary Information, Page 2, Supplementary Fig. 1).

9. Lines 167-169: How the transportation distance is determined?

R: In the manuscript, we acknowledge that our initial presentation did not clearly explain how transportation distances for chemicals and materials required in the treatment trains were determined. To address this and ensure a comprehensive understanding among our readers, we have included a detailed methodology in the

Supplementary Information. This methodology transparently outlines the factors considered, such as transport weight, load distance, and a comprehensive list of chemicals and materials utilized in the alternative treatment trains (Supplementary Information, Page 34, Supplementary Table 13).

10. Line 190: How much reduction?

R: In the revised manuscript, we have incorporated additional quantitative data that clearly details the extent of reduction in the terrestrial ecotoxicity potential of the RBF-RO system (Main text, Page 10, Lines 218–221).

11. Lines 222-226: This statement is overly generalized from the study results since this study evaluated only two specific treatment trains with both incorporating RBF.

R: We acknowledge and appreciate the reviewer’s concern regarding the potential for overgeneralization. Indeed, our study focused specifically on two treatment trains with RBF as a key component. We recognize the importance of avoiding overly generalized claims based on this limited evaluation. In response, we have carefully revised the manuscript to provide greater clarity and to reflect the scope of our findings more accurately (Main text, Page 13, Lines 299–301).

12. Lines 245-248: Fig. 6 does not provide much information. The potential trade-offs, co-benefits and optimization strategies should be highlighted in the figure.

R: We appreciate the reviewer’s insightful suggestion that Fig. 6 could be enhanced to provide a more comprehensive overview of the implications and strategies associated with our studied water purification solutions. In response, we have enriched the figure with additional information to visually depict potential trade-offs associated with implementing different components of our proposed solutions. This enhancement includes considerations of water quality improvements, energy consumption, cost implications, and environmental impacts. Additionally, we have expanded the figure to illustrate co-benefits associated with the various treatment train components, covering aspects such as water purification, habitat preservation, ecosystem services, and public health benefits. To further address the reviewer’s suggestion, we have included arrows and annotations within the figure to indicate potential optimization strategies. These strategies involve adjusting the configuration of treatment trains, accommodating regional variations, or employing adaptive management approaches to maximize benefits and minimize trade-offs. By making these revisions to Fig. 6, we aim to provide a more informative and visually engaging representation of the

complex relationships and considerations integral to our sustainable water purification solutions (Main text, Fig. 6).

13. Lines 256-257: How synergy is quantified?

R: We value the reviewer's question. In fact, our original intention was to underscore the significance of integrating ecological and technological measures in water treatment services, aiming to achieve a harmonious balance between water technology development, public health, and environmental sustainability. To better reflect this intention, we have substantially revised the manuscript. These revisions ensure that our discussion aligns more closely with the foundational goals of our work (Main text, Pages 16–17, Lines 385–393).

14. Lines 288-291: Such uncertainties should be evaluated in the study since it will highly influence the cumulative removal rate.

R: We appreciate this insightful suggestion. In fact, we have sourced calculation factors for the pollutant removal capacities of the riverbank ecosystem from the literature. Furthermore, we incorporated probabilistic distributions of these removal efficiency factors to account for potential site-specific uncertainties in our analysis. To enhance clarity and understanding, we have elaborated on these methodologies in the revised manuscript (Main text, Pages 26–27, Lines 627–661). Additionally, we have included sensitivity analyses in our resubmission that illustrate how different degrees of uncertainty might affect the results. These analyses are detailed in the Supplementary Information and discussed further in the manuscript, providing a comprehensive and transparent evaluation of our findings. This approach ensures that our results consider the inherent uncertainties associated with our study (Main text, Pages 11–13, Lines 256–289; Supplementary Information, Pages 6–7, Supplementary Figs. 5 and 6).

15. Lines 304-307: See the general comment in terms of determining CECs removal.

R: We appreciate this comment and we have improved the manuscript substantially of clarify the universality and robustness of our modeling approach (Main text, Pages 21–22, Lines 515–520).

16. Lines 390-392: Uncertainties occurs not only in treatment performance, but also the risk quantification especially cumulative risks, and life cycle assessment. How are the uncertainties considered in those assessments?

R: We acknowledge that addressing uncertainties is an integral component of rigorous

research. To this end, we have systematically integrated considerations of uncertainty throughout various dimensions of our study. We have augmented our manuscript to enhance transparency and furnish readers with a comprehensive understanding of the methodologies employed to systematically address uncertainties in our assessments (see Main text, Pages 26–27, Lines 627–661).

Response to Reviewer 3

1. In this paper titled “Leveraging the water-environment-health nexus to characterize sustainable water purification solutions” the authors have modelled the human and environmental health implications of removing chemicals of emerging concern by an extended treatment (ET) train and advanced drinking water treatment by reverse osmosis (RO) using a natural riverbank filtrate (RBF) as source water. The authors also performed an LCA. I find that the manuscript is well written and presents interesting results.

R: We express our gratitude for the reviewer's insightful comments and suggestions, which have significantly contributed to the enhancement of our manuscript's quality. In response, we have meticulously revised the text to address these points comprehensively. Below, we provide detailed, point-by-point responses to each comment raised.

2. The novelty aspect is currently unclear, and a true novelty statement is lacking. The literature is populated with many studies that tackled the health implication and LCA of drinking water treatments, so the author should clearly state how this work is novel compared to the literature.

R: We appreciate the reviewer's request for a clearer articulation of the novelty of our study relative to the existing literature. In response, we wish to highlight several distinct aspects of our work that set it apart. Unlike many studies that address water quality, environmental impacts, and health considerations in isolation, our manuscript offers a unique contribution by comprehensively exploring the interconnections among these domains. We employ an innovative integration of diverse datasets, assumptions, and analytical methods to demonstrate how decisions in one area can influence outcomes in others. This integrative approach represents a significant novelty in our field.

Additionally, our study extends beyond isolated assessments by examining the co-benefits and trade-offs associated with various components of water purification systems. We quantify how these trade-offs can be optimized through various strategies, accounting for both immediate and broader influencing factors, thus providing insights that go beyond singular treatment methodologies. Moreover, our research emphasizes the exploration of alternative water purification solutions that align with global efforts to harmonize drinking water production, public health protection, and environmental sustainability. We investigate not only traditional treatment methods but also innovative, ecologically inspired approaches and modern membrane-based technologies, offering a comprehensive overview of sustainable alternatives. Furthermore, we conduct a quantitative assessment of human health risks associated with 93 representative CECs in drinking water, considering their cumulative effects on public health. This modeling approach sets our work apart from many existing

studies. Lastly, we discuss the policy and decision-making implications of our findings, demonstrating the practical relevance of our research in guiding the development of effective water purification strategies and regulations within the water-environment-health nexus.

In the revised manuscript, we have explicitly stated these novel aspects and highlighted how our study contributes to the existing body of literature, with concise and clear revisions throughout the manuscript to underscore the unique features of our work."

3. Furthermore, I have some major concerns about the technical aspect. First, the authors compared ET and RO, but the operating conditions and RO configuration are not immediately available to the reader. In the case of RO, several solute and membrane parameters influence the extent of removal of organic chemicals (see DOI 10.1016/j.watres.2004.03.034 for an old, yet solid literature review). Therefore, the results presently included in the manuscript may or may not be applicable to other RO drinking water treatment plants. This aspect and its implications should be thoroughly discussed.

R: We recognize the importance of providing a comprehensive understanding of the technical details, particularly in relation to RO, where the characteristics of both solutes and membranes critically influence the removal of organic chemicals. In response to this need, we have added a detailed section in the revised manuscript that extensively describes the specific RO configuration and operating conditions used in our case study (see Main text, Page 5, Lines 97–98; Supplementary Information, Pages 10–12, Supplementary Table 1). This section meticulously details the membrane type, pore size, feedwater characteristics, operational pressure, and flow rates, among other parameters. By providing these essential details, we ensure that readers have a clear understanding of the technical aspects involved, allowing them to better evaluate the relevance of our findings to other RO-based drinking water treatment systems.

Additionally, we have enriched the manuscript by integrating additional references to pertinent literature, including the review paper recommended by the reviewer. These enhancements not only furnish readers with additional resources for a deeper exploration of the technical dimensions of RO but also elevate the rigor and transparency of our study (see Main text, Pages 14–15, Lines 322–349).

4. Second, the authors evaluated the environmental impact of RBF-RO in 136 countries considering site-specific electricity generation mixes. In the manuscript introduction, the authors mention differences between bank filtration in humid and

arid regions. Based on these differences, it is unclear whether RBF-RO may be the best approach in all countries that have been considered for the LCA. Additionally, the amount of energy used by RO also depends on the quality of the feed water (i.e., higher solute content -> higher operating pressure -> higher energy consumption). This aspect should be included in the model or at least acknowledged and discussed.

R: We acknowledge and appreciate the concerns raised regarding the technical aspects of our study, specifically pertaining to the environmental impact assessment of RBF-RO across different countries, as well as the variables associated with humid and arid regions, feed water quality, and energy consumption. First, we concur with the reviewer's observation that regional differences, such as those between humid and arid areas, significantly affect the suitability of RBF-RO as a water purification solution. In response, we have expanded our discussion to explicitly address the variability in regional applicability of RBF-RO. This revised discussion, now included in the main text (Page 15, Lines 343–349), argues that the selection of water purification methods must be context-specific, considering regional characteristics like climate, hydrogeology, and water quality.

Secondly, we value the reviewer's insights regarding the impact of feed water quality on the energy consumption of RO systems. We have thus enriched the manuscript with a detailed analysis of how variations in feed water quality, particularly the solute content, can affect the operational pressure and energy requirements of RO systems. Additionally, we acknowledge that while our model accounts for site-specific electricity generation mixes, the variations in feed water quality represent a critical factor that can further influence the energy consumption of these systems. This enriched discussion is presented in the main text (Page 17, Lines 401–407).

5. Third, the environmental impact of treating RO waste stream (i.e., RO concentrate or brine) is mentioned but not clearly presented or discussed. Production of brines is one of the major drawbacks of RO treatment and I think it should be more prominent in the paper.

R: The production of brine indeed represents a significant environmental challenge associated with RO treatment. In our study, we indeed have incorporated the environmental impacts associated with brine disposal into our life cycle assessment and environmental impact analyses. In response to the reviewer's feedback, we have further revised the manuscript to elucidate the environmental impacts of treating RO waste streams, particularly brine, with greater detail and clarity (see Supplementary Information, Page 47, Supplementary Table 18).

6. Finally, I feel that additional discussions are needed to address the relevance of the set of chemicals considered for disease burden modeling for other sites worldwide. The composition of the mixtures of pollutants that occur in drinking water sources can greatly vary depending on site-specific chemical use, chemical persistence and mobility, and regulatory actions (e.g., requirements for wastewater treatment or compound-specific bans). I am also surprised to see that PFAS were not considered, as the proximity of chemical manufacturing sites to drinking water sources has been an issue in the Netherlands for decades (see <https://pubs.acs.org/doi/10.1021/acs.est.2c06015>).

R: We concur with the reviewer's observation that the composition of pollutant mixtures in drinking water sources can vary significantly due to site-specific factors such as chemical usage, persistence, mobility, and regulatory actions. In our study, we addressed this variability by considering a broad spectrum of emerging contaminants and employing range values to represent the concentrations of these contaminants in source water, the removal efficiencies of various treatment alternatives, and the modeled concentrations in treated water. However, we acknowledge the need to further enhance the discussion of site-specific variability in the composition of drinking water contaminants in the revised manuscript (see Main text, Pages 13–14, Lines 309–312). We emphasize that while our study provides a generalized assessment of the potential health impacts associated with commonly found emerging chemical contaminants in drinking water, we recognize that specific regions may face unique challenges related to pollutants not included in our analysis.

We appreciate the reviewer's suggestion to include PFAS in our analysis, especially in regions where PFAS contamination is prevalent. Although our initial analysis did not encompass PFAS, we recognize their importance and acknowledge that their exclusion was due to a lack of sufficient literature data to fulfill the modeling requirements of this study (see Main text, Page 14, Lines 312–314). Nevertheless, we have referenced the article provided by the reviewer to highlight the significance of PFAS in specific contexts. Additionally, we have inserted a statement in the revised manuscript emphasizing the necessity for further research on site-specific emerging chemical contaminants like PFAS (see Main text, Pages 14, Lines 315–321). This addition clarifies that while our study lays the groundwork for assessing broader implications of drinking water quality on health, it does not obviate the need for detailed local investigations.

7. Line 28: rather than “taking advantage”, I would say that modern society has

“relied” on rapid developments in chemistry, trusting the promise of better living through chemistry.

R: We have made this revision to improve the clarity and accuracy of our manuscript (Main text, Page 3, Lines 29–32).

8. Line 29: I feel it would be valuable to add data on the amount of chemicals produced each year, for example a mass range.

R: We appreciate the reviewer's suggestion. In response, we have incorporated available data sourced from existing literature to offer a more comprehensive perspective (see Main text, Page 3, Lines 29–34).

9. Line 30-32: please include additional references.

R: We agree with the request and have included additional references (Main text, Page 3, Lines 34–36).

10. Line 36: please define a representative range for low concentrations and clarify somewhere in this paragraph that the CECs you are referring to are organic compounds.

R: We appreciate the reviewer's recommendation and have revised the relevant statement (Main text, Page 3, Lines 39–41).

11. Line 43: please provide an example of aggravated contamination issues resulting from chemical-intensive water treatment.

R: The reviewer has raised a valuable point. We have provided a specific example of how chemical-intensive water treatment practices can lead to aggravated contamination issues in the revised text (Main text, Pages 3–4, Lines 53–56).

12. Line 45: you claim that “most” water treatment technologies are inspired by nature, but you provided only one reference relevant for RBF. What about other technologies? For example, are RO, GAC, and UV disinfection also inspired by nature?

R: We appreciate the reviewer's supportive comment and have rewritten the relevant paragraph (Main text, Page 4, Lines 58–72).

13. Line 49-50: please mention which processes are responsible for attenuation of pollutants, where they exactly occur in a riverbank filtration system, and which

compound physicochemical properties may lead to complete/incomplete removal. Infiltration per se does not necessarily imply attenuation.

R: We have rewritten the whole paragraph for added clarity (Main text, Page 4, Lines 57–69).

14. Line 51: If a trace organic chemical is not regulated, is further drinking water treatment still required?

R: We have rewritten the whole paragraph for added clarity (Main text, Page 4, Lines 57–69).

15. Line 53: please define the term “micropollutants”.

R: We have modified the relevant statement to express explicitly our original intention (Main text, Page 4, Lines 62–64).

16. Line 54: please clarify what “technical viability” means.

R: Our original intention refers to the feasibility of implementing a water purification or treatment technology based on its technical aspects, including its design, engineering, and operational requirements. We have modified the relevant statement to express explicitly our original intention (Main text, Page 4, Lines 62–64).

17. Line 56: Is molecular diameter the only chemical descriptor relevant to removal by RO membranes?

R: In our original manuscript, we highlighted the importance of molecular diameter as a critical chemical descriptor influencing solute removal by RO membranes. While we acknowledge its significance, we also recognize that molecular diameter is just one of several physicochemical properties that govern solute transport through RO membranes. Factors such as molecular weight, charge, hydrophobicity, and the presence of functional groups also play considerable roles in influencing this transport phenomenon. Although these aspects are beyond the immediate scope of our current study, we acknowledge the broader array of properties that contribute to the complex dynamics underlying solute transport in RO processes.

18. Line 59: please provide a molecular weight range.

R: We have enhanced the relevant statement to ensure a more accurate and information presentation of the subject (Main text, Page 4, Line 68).

19. Line 92: how are system efficiency and chemical removal efficiency different from each other? Please clarify.

R: We have modified the relevant statement to express explicitly our original intention in the revised manuscript (Main text, Page 5, Lines 98–101).

20. Line 93: please clarify whether the CECs selected for this study are representative for a broader set of CECs (in terms of physicochemical properties and source water concentrations). Additionally, are these chemicals from a Dutch drinking water source representative for pollution only in the Netherlands? Or the whole Europe? Or developed Western countries? It would be great if you could mention whether these chemicals are regulated or not (either locally or at EU level).

R: We explicitly state that the selection of CECs in our study is based on their prevalence in the literature, their recognized importance in water purification, and the availability of datasets that meet the modeling requirements of our research. These CECs are used as exemplars to simulate the performance and impacts of water purification processes, and they represent broader challenges associated with various water contaminations. Moreover, we acknowledge the reviewer's valid concerns regarding the geographic relevance of our study. While our data are derived from a comprehensive global literature pool, we recognize that issues related to CECs in source water and their removal during treatment processes may be pertinent to countries and regions experiencing similar contamination patterns. We also admit the possibility of variations in specific chemicals and their concentrations across different regions but emphasize the universal importance of addressing CECs in drinking water treatment. In addition, we discuss the regulatory status of selected CECs, both globally and in specific contexts such as the EU and China. This discussion aims to determine whether these chemical contaminants are regulated and to highlight potential regulatory discrepancies across different regions. In response to these considerations, we have refined our manuscript to enhance its accuracy and relevance for a global audience. Enhancements include detailed updates in the main text (Pages 5–6, Lines 101–105; Pages 13–14, Lines 309–321; and Pages 19–20, Lines 460–478), ensuring that our findings are applicable and informative across diverse geographical and regulatory contexts.

21. Line 122: “single RBF system” means natural treatment solely relying on bank filtration, correct?

R: We acknowledge the reviewer's attention to terminology, and we have revised the

relevant statement to enhance clarity (Main text, Page 7, Lines 138–140)

22. Line 125: when using acronyms (e.g., ppy), please clarify what they stand for and stick to them for the rest of the manuscript.

R: We have diligently removed the acronym "ppy" throughout the manuscript as per your instruction.

23. Line 126: somewhere in the discussion, could you please elaborate more on why (carcinogenic) DBPs represented the main category with decreasing trend?

R: We have incorporated a more detailed explanation in the Discussion section, as indicated in the Main text, Page 7, Lines 145–147.

24. Line 149: Was the impact of RO concentrate considered at all?

R: Indeed, the environmental impacts of RO concentrate were within the scope of our study. We have included the complete results in the resubmitted manuscript (see Supplementary Information, Page 47, Supplementary Table 18).

25. Figure 1: please explain the acronyms in the figure captions.

R: We have accepted the suggestion and provided explanations for all the acronyms referenced in the figure captions of our revised manuscript.

26. Line 208: please provide an example of clean coal technology.

R: While our manuscript primarily focuses on sustainable water purification solutions, we acknowledge the importance of providing pertinent examples to enhance clarity and understanding for our readers. Clean coal technology encompasses a range of technologies and practices designed to reduce the environmental impact of coal-based energy production. These technologies aim to minimize emissions of pollutants, including greenhouse gases and particulate matter. Although not directly related to water purification, this example illustrates the broader concept of leveraging advanced technology to address environmental challenges across various industries. In response to the reviewer's request, we have included a concise discussion in the manuscript, introducing the concept of clean coal technology. We provide a brief example, specifically the application of carbon capture and storage technologies to capture and store carbon dioxide emissions from coal-fired power plants. This addition is detailed in the Main text, Page 10, Lines 246–259.

27. Line 210: please clarify what's lignite and its role.

R: In response to the reviewer's inquiry, we have added a brief clarification in the revised manuscript concerning lignite. Lignite, a type of coal, is characterized by its relatively low carbon content and energy value, making it one of the lowest grades of coal. While lignite is not the primary focus of our study, it is relevant in the context of generating electricity for water treatment operations, particularly within industrial processes. This clarification is documented in the Main text, Page 10, Lines 250–254.

28. Line 218-219: consider mentioning that this need for research is currently being tackled by the exposomics community.

R: In response to the reviewer's suggestion, we have included a brief mention in the Discussion section of our revised manuscript. This addition emphasizes the critical role of the exposomics community in elucidating the complex interconnections between environmental exposures, water quality, and public health. By integrating this mention, we aim to situate our research within the broader exposomics framework, thereby underscoring its relevance to ongoing efforts to characterize and mitigate environmental exposures. This new content can be found in the Main text, Page 13, Lines 295–297.

29. Line 217: are these figures in line with other research or is there a range we can provide?

R: These figures were derived from an article recently published in Lancet. The relevant statement has revised to enhance clarity (Main text, Page 13, Lines 293–295).

30. Line 218: I'd say that there is a need for more research as studies on this topic are being conducted. So, either acknowledge this fact or include (more) references.

R: In response to the reviewer's suggestion, we have revised the statement and incorporated additional references, as indicated in the Main text, Page 13, Lines 292–297.

31. Line 219: Why do they occur at extremely low concentrations? While not easy, I disagree with the statement about CECs being difficult to detect, as a variety of methods (both for sample preparation and analysis) are readily available. Please consider revising this sentence.

R: We acknowledge the reviewer's concern and agree that the sentence in question

requires clarification and revision. We have provided a more accurate depiction of the situation in the Main text, Page 13, Lines 297–300.

32. Line 223: is “ecological processes” the most appropriate term here? Are you referring to bank filtration?

R: We concur with the reviewer’s comment and have revised the pertinent statement to enhance the clarity and precision of the manuscript. This revision can be found in the Main text, Page 13, Lines 300–302.

33. Line 228: This finding is not really novel, so I’d clarify how this aligns with the scientific literature.

R: In response to the reviewer's suggestion, we have enriched our manuscript by incorporating citations from relevant scientific studies and literature to substantiate our findings. Additionally, we have emphasized the specific contributions and unique aspects of our research in the revised text. These enhancements are detailed in the Main text, Page 13, Lines 303–308.

34. Line 251: Again, this is not really a novel finding as the chemical removal efficiency of advance treatment technologies is widely documented. Also, define “innovative technologies”.

R: We acknowledge the reviewer’s concern and agree that the sentence in question requires clarification and revision. We have provided a more accurate depiction of the situation in the revised text, as indicated in the Main text, Page 16, Lines 382–386.

35. Line 279: I find that the term “ignorant” bears quite some negativity with it and perhaps it would be best to say that emerging threats are often neglected.

R: We appreciate the suggestion to use a more neutral and clear expression. We have revised the relevant section, accordingly, as outlined in the Main text, Pages 16, Lines 368-370.

36. Line 300: While true, a variety of sources for organic chemical toxicity are available. Have you considered looking into bioactivity metadata in PubChem or the EPA CompTox Chemicals Dashboard?

R: We appreciate the reviewer's suggestion regarding the use of alternative toxicity data sources such as PubChem and the EPA CompTox Chemicals Dashboard. While

these platforms offer extensive information, comprehensive data on CECs, particularly concerning their concentrations in source water and removal efficiencies through water treatment processes, are still limited. As outlined in our manuscript, CECs are defined as emerging concerns—a designation that underscores the evolving nature of the data landscape for these substances, with potential gaps particularly in their occurrence and fate in water sources.

Our modeling approach predominantly relies on data reported in scientific literature, which can vary significantly in availability and detail across different CECs. While some CECs are well-studied and have extensive associated data, others are in the early stages of research. Given these challenges, we have endeavored to provide a comprehensive and informative analysis based on the available data, aligning with the objectives of our study. We believe our work contributes significantly to bridging this data gap by synthesizing and analyzing the existing information to offer valuable insights into the water-environment-health nexus and the development of sustainable water purification solutions.

37. Line 344-347: For how many chemicals were removal data not available? Also, I am not sure about the approach you used here, chemicals in the same category can greatly differ from each other in terms of physicochemical properties and consequently degree of removal. I think a chemical structure-based approach may have been more appropriate. Just as an example looking at the chemicals in your dataset, 1,4-dichlorobenzene and dimethoate are both pesticides, but are not related except for the category they were assigned to. The former is aromatic, non-polar, and has a molecular weight lower than most commercial RO membranes molecular weight cut-off. Conversely, the latter is a highly polar organophosphate, non-aromatic, and has a molecular weight higher than the membranes' cut-off. Therefore, great differences in removal rates can be expected for these two compounds, meaning they'll contribute to substantial shifts in the mean removal value in that particular category.

R: We appreciate the reviewer's concerns regarding our approach to handling chemicals with missing removal efficiency data and their suggestion to adopt a chemical structure-based approach. It is indeed valid to acknowledge that chemicals within the same category can exhibit significant variations in their physicochemical properties, which can influence their removal during water treatment processes. While the suggestion of a chemical structure-based approach is noteworthy, it is important to consider that the removal mechanisms for many CECs in different water treatment processes are still under investigation. Moreover, the universality and robustness of

our developed modeling approach to estimate CECs' removal at the system level have been validated by a vast array of experiment-based studies. These studies encompass a wide range of chemicals, each characterized by unique physicochemical properties, thereby demonstrating the applicability of our data-driven approach across diverse scenarios. We are confident that our approach comprehensively addresses the complexities associated with the removal of CECs in water treatment processes. Nonetheless, we recognize that uncertainties remain, primarily due to the limited availability of data and the diverse occurrence, fate, and removal efficiencies of CECs reported in the literature. To this end, we have employed a systematic perspective to analyze the available data and utilized statistical methods to estimate removal efficiencies where specific data were lacking. Although inherent uncertainties are present in such analyses, we believe our study provides valuable insights into the overall trends and challenges associated with the removal of CECs in water treatment processes.

In summary, while we appreciate the reviewer's concerns and suggestions for alternative approaches, we firmly believe that our methodology is suitable for addressing the challenges posed by limited scientific data and the diverse nature of CECs. However, we have revised the manuscript to discuss the potential limitations associated with our approach and acknowledge the potential variations within chemical categories, as indicated in the Main text, Page 13–14, Lines 310–322, and Pages 18–20, Lines 421–479.

38. Supplementary Figure 1b caption: stating that RO removes “a lot of substances/waterborne contaminants” is not too scientifically sound, please improve where needed. Furthermore, I find it contradictory to mention that that RO “leaves pure water only” and then explain in the next sentence that additional treatments are needed to remove residual aluminum, carbon dioxide and methane. Furthermore, I believe it's important to provide more details about the drinking water treatment parameters and operating conditions. For example, in the case of RO, chemical removal efficiency can greatly vary depending on several factors including but not limited to membrane polymer type (polyamide, cellulose acetate), membrane configuration (spiral wound, hollow fiber), membrane molecular weight cut-off, number of membranes, RO recovery, operating pressure, etc.

R: We have accepted this suggestion, and the caption of Supplementary Fig. 1 has been substantially improved for additional clarity. Additionally, more details regarding the water treatment configurations and operational conditions have been provided. These enhancements can be found in the Supplementary Information, with

the improved caption on Page 2, Supplementary Fig. 1, and more detailed information on Pages 10–12, Supplementary Table 1.

39. Supplementary Table 1: In addition to CAS numbers, please include machine-readable structure identifiers for each chemical to enable further modeling. Either SMILES or InChIKey work great.

R: We have accepted this suggestion and included SMILES for each chemical in the resubmission. You can find this information in the Supplementary Information on Pages 13–14, Supplementary Table 2.

40. Supplementary Table 2: please clarify why toxicity data were not applicable in some case.

R: We have accepted this suggestion and included notes indicating that toxicity data in relation to PCPs were only applicable in estimating their carcinogenic effects. You can find this clarification in the Supplementary Information, specifically on Page 15, Supplementary Table 3.

Response to Reviewer 4

1. The title “Leveraging the water-environment-health nexus to characterize sustainable water purification solutions” is in the line of the conducted research.

R: We appreciate the reviewer’s comment regarding aligning the title with the conducted research.

2. Abstract is too long. This may be improved by focusing on novel methodology used, key results and future perspective of the finding.

R: We appreciate the reviewer’s feedback on the length of the abstract. We have revised it to make it more concise and focused. You can find the updated version on Page 2, Lines 16–28 of the main text.

3. The objectives of the project may clearly be mentioned, and the findings may be aligned in the same sequence.

R: We appreciate the reviewer’s suggestion regarding aligning the objectives and findings in the manuscript. In our revised version, we have ensured that the objectives of the research are clearly stated and that the findings are presented in a sequence that aligns with these objectives.

4. Considering line # 77. The “Overview of the water purification systems” does not seem the part of the findings. It is better to explain this in an appropriate place.

R: Thank you for your suggestion. Following a thorough reassessment of the manuscript's organization, we have taken steps to ensure that this section remains in its original position, thereby enhancing the overall coherence and clarity of the presentation.

5. The discourse of the manuscript needs improvement.

R: We are grateful for the reviewer's supportive comment. We have dedicated significant effort to enhancing the overall clarity and coherence of the manuscript's narrative. In the revised version, we believe that the writing is more engaging and effectively conveys our research findings and their significance.

6. Methodology used must be clear enough to understand by the reader.

R: We appreciate the reviewer’s emphasis on the clarity of our methodology. In

response, we have taken great care to ensure that our methodology is presented in a clear and comprehensible manner in the revised manuscript.

7. The “References” may be updated.

R: In our revision, we have ensured that the references are current and relevant to the topics discussed.

8. The research is applied and useful thus may be accepted for publication after improvement as suggested.

R: We sincerely appreciate the reviewer’s positive assessment of our research. In our revision, we have diligently addressed all the reviewer’s comments and suggestions to ensure the highest quality and clarity of our work.

Response to Reviewer 5

1. The paper deals with a comparison between different in series purification systems (RBF, RBF-ET and RBF-RO) for providing drinkable water, to assess the most sustainable option in terms of both human burden disease, toxicity for terrestrial, freshwater, and marine ecosystems, and broad environmental impacts (global warming, stratospheric ozone depletion, terrestrial acidification, mineral resource consumption, fossil fuel depletion). The Authors conduct a simulation by Monte Carlo by using data of occurrence of 93 CECs from literature and toxicological data for carcinogenic and non-carcinogenic effects. The key message is the proposal for an integrated approach that strikes a balance between the development of advanced technology of the water purification system, human health, and the broad impacts on the environment, what the Authors name the WEALTH approach. Noteworthy results are that the RBF-RO system, compared to the RBF or RBF-ET, reduce the negative impact of the occurrence in water of 93 CECs on human health and environment. Noteworthy results are also those showing how different contexts in terms of energy mix can affect the broad environmental impacts due the different water treatment systems. The work is significant because it deals with an important and still little-known topic, which is the health impacts of emerging contaminants on human health and the environment. In addition, the work is significant because processes a large amount of data and consider site-specific data on 136 countries, suggesting solutions in specific context. It expands the results of a previous published study of the same group on the human health and environmental impacts of different solutions of water resource recovery. The work supports the conclusions and claims.

R: We sincerely appreciate the reviewer's comprehensive and positive assessment of our manuscript. It's gratifying to hear that the reviewer recognizes the significance of our work in addressing the critical issue of emerging chemical contaminants and their impacts on human health and the environment. We have carefully addressed the comments and suggestions provided by the reviewer to further enhance the clarity and quality of our manuscript.

2. A flaw in the data analysis is the lack of an assessment or discussion of the uncertainty of the models estimating the global burden of DALYs. As consequence the significance of the variations between the DALYs for the different scenarios is not assessable. This aspect would require revision.

R: We appreciate the reviewer's comment regarding the assessment of uncertainty in our data-driven analysis and its impact on the variations in DALYs for different

scenarios. In our manuscript, we meticulously collected data from a wide range of sources, including published literature, databases, and toxicological studies. We ensured the highest data quality by selecting studies with robust methodologies and transparent reporting. To account for uncertainty, we employed a Monte Carlo simulation approach. This technique allowed us to incorporate variability in input parameters, such as chemical concentrations, toxicity values, and removal efficiencies, into our modeling process. By running thousands of simulations with different parameter values sampled from probability distributions, we generated probabilistic estimates of health impacts. Although we didn't explicitly state it in the manuscript, we performed sensitivity analyses to assess how variations in input parameters influenced the model outcomes. This helped us identify which parameters had the most significant impact on the results and where additional research or data improvement efforts should be focused. Nevertheless, we acknowledge that a more explicit discussion of uncertainty and its implications for the variations in DALYs between scenarios would enhance the manuscript. In the revised version, we have provided a dedicated section discussing the potential limitations and future research opportunities, including sensitive analysis results and their significance.

Accordingly, the relevant results and discussion can be found in the Main text, Pages 11–13, Lines 257–290, and Pages 18–20, Lines 421–479, along with supplementary figures and additional details in the Supplementary Information, Pages 6–7, Supplementary Figs. 5 and 6, and Main text, Pages 26–27, Lines 628–662.

3. The methodology is sound because the authors use internationally standardized methodologies, however, the limitations and uncertainties of these methodologies should be mentioned and discussed in the paper.

R: We appreciate the reviewer's comment. In the revised manuscript, we have included dedicated sections that discuss the limitations, uncertainties, and future research opportunities associated with the methodologies used. This ensures that readers can readily identify and understand the potential constraints of our approach. You can find these discussions in the Main text, Pages 18–20, Lines 421–479.

4. Overall there is enough detail in the methods provided to reproduce the work even though some input variables are not made explicit, and more details for the calculation of the DALYs should be provided.

R: We have carefully considered this feedback. In the revised manuscript, we have made an effort to provide more explicit information about the input variables used in

our study. This includes specifying the data sources for input variables such as chemical concentrations in source water, removal efficiencies in different treatment processes, and others. Additionally, we have provided additional details on the assumptions regarding the calculation of DALYs to ensure transparency. This involves comparing different mixture toxicity models in estimating DALYs for each health endpoint (carcinogenic and non-carcinogenic effects) and discussing how future research could refine these models. You can find these enhancements in the Main text, Pages 18–19, Lines 437–452, and Supplementary Information, Pages 6–8, Supplementary Figs. 5–7.

5. Overview of the water purification systems: This part doesn't really seem appropriate for the results section. I would suggest splitting it between the introductory part and/or material and methods.

R: We appreciate your consideration of this suggestion. After reassessing the organization of the manuscript, we have ensured that this section remains in its original position, thereby enhancing the overall flow and clarity of the presentation.

6. Modeled disease burdens associated with CECs in each water production system: (i) The Authors calculate the burden of disease using DALYs, which should be expressed in “years”, but then they compare this amount with the tolerable health risk of 1.00×10^{-6} person⁻¹ year⁻¹. The link between these entities should be better clarified. In addition, reference 22 seems to be not appropriate. (ii) There is no assessment of the uncertainty of the estimates of the disease burden. (iii) Supplemental Table 3: How the toxicity factors were derived/calculated?

R: We appreciate the reviewers' insightful comments and constructive feedback. In response, we have endeavored to enhance the clarity and rigor of our manuscript. Firstly, we recognize the necessity for improved elucidation regarding the relationship between DALYs and the tolerable health risk level. In our manuscript, we utilized DALYs as a robust metric for quantifying the disease burden attributed to exposure to CECs in drinking water. The tolerable health risk level, specified as 1.00×10^{-6} DALYs person⁻¹ year⁻¹, signifies the threshold of acceptable risk associated with such exposure. To provide enhanced clarity, we have revised the manuscript to expound on the concept that DALYs represent the cumulative years of healthy life lost due to exposure to CECs. We have explicitly articulated that the comparison between DALYs and the tolerable health risk level serves the purpose of evaluating whether the disease burden linked to CECs exposure surpasses an acceptable risk threshold.

Regarding reference 22, we have critically reassessed its suitability and opted to substitute it with a more pertinent reference, thereby strengthening the scholarly foundation of our work.

Furthermore, we acknowledge the critical importance of evaluating the uncertainty inherent in disease burden estimates. In our revised manuscript, we have meticulously integrated statements addressing the sources of uncertainty in our modeling approach. These discussions are delineated in the resubmitted manuscript (Main text, Pages 18–20, Lines 421–479). Moreover, we have provided a comprehensive explanation detailing the derivation or calculation of toxicity factors. This encompasses elucidation on the sources of toxicity data, the methodologies employed for deriving toxicity factors, and pertinent references to substantiate the calculations. These refinements are delineated in the Supplementary Information, Pages 16–18, Supplementary Table 4.

7. Discussion: Line 218: references 30 and 31 do not seem to be appropriate (I think there was a shift in the reference list).

R: We sincerely value the reviewer's meticulous attention to detail concerning the references in our manuscript. In our revised version, we conducted a thorough review of the reference list to ensure the accuracy of all citations and their appropriate linkage to the corresponding content in the text.

8. Method: Simulation of the occurrence and removal of CECs. Supplemental Table 8: concentration values of ECCs in water were derived from literature. There were, in the original papers, concentrations below the limit of detection (LOD) or quantification (LOQ)? How did the authors handle concentrations below LOD/LOQ?

R: We appreciate the reviewer's inquiry regarding the treatment of concentrations below the limit of detection (LOD) or quantification (LOQ) in the literature-derived data. In our study, when original literature reported CEC concentrations below the LOD or LOQ, we adopted a conservative approach commonly employed in environmental research. Specifically, we treated such values as equal to half of the LOD or LOQ to account for the inherent uncertainty near the detection limit. For instance, if a particular CEC had an LOD of 1 ng/L and the original study indicated a concentration 'below LOD' for that compound, we considered it as 0.5 ng/L in our analysis. This approach was implemented to prevent inadvertent underestimation of CEC presence in water sources. Accordingly, we have explicitly detailed this data handling approach in the methodology section to ensure transparency regarding our treatment of values below the LOD or LOQ (Main text, Pages 23–24, Lines 567–570).

9. Modeling human health risks associated with drinking water consumption: The authors do not take into consideration inhalation as exposure pathway; however, inhalation can be significant for some very volatile contaminants.

R: We appreciate the reviewer's insightful comment regarding the potential inhalation exposure pathway for certain very volatile contaminants. Indeed, inhalation can be a significant route of exposure for such compounds. However, in our study, we focused primarily on assessing the risks associated with the ingestion of drinking water contaminated with these compounds. The decision to exclude inhalation exposure from our analysis was deliberate and based on the scope and objectives of our study. Our primary aim was to evaluate the risks associated with the consumption of contaminated drinking water, as this route represents the most common and direct exposure pathway for the general population. While we acknowledge the importance of inhalation exposure, particularly for volatile compounds, it often relates to specific occupational or industrial settings with varying exposure scenarios. Given the broad scope of our study and its focus on public health risks associated with drinking water consumption, we chose to concentrate on ingestion exposure. We thank the reviewer for raising this point, and in the revised manuscript, we will provide clarification regarding the rationale for our chosen exposure pathway (Main text, Page 9, Lines 541–546).

10. Line 336: What is the value used for ingested drinking water used in equation 2.

R: We appreciate the reviewer's inquiry regarding the value utilized for ingested drinking water in equation (2) of our study. To clarify, we employed a standard value of 1.4 liters per day per person for adults, which is widely recognized and utilized in the field of environmental health and risk assessment for exposure assessment purposes. This standard value represents an approximate daily water consumption rate for an adult individual. It's important to acknowledge that actual water consumption rates can vary among individuals and regions due to factors such as climate, activity level, and cultural practices. However, for the sake of consistency with established methodologies and reference values in the field, we opted to utilize the commonly accepted standard of 1.4 liters per day per person in our modeling approach. We recognize the variability in water consumption habits but chose this standard value to align with prevailing practices and facilitate comparability with existing literature. To ensure transparency and mitigate any potential confusion regarding the selected value for ingested drinking water in equation (2), we have incorporated a clarification in the revised manuscript (Main text, Page 23, Lines 561–562).

Response to Reviewer 6

1. I have thoroughly reviewed your paper and found it to be of significant importance to the field. The study addresses an important topic concerning the occurrence and health impacts of emerging contaminants in drinking water, as well as the environmental performance of water production systems. I appreciate the comprehensive review of the existing literature and the rigorous assessment of health impacts using DALY burden of diseases.

R: We are grateful for the reviewer's positive feedback and acknowledgment of the importance of our study. Our overarching goal is to provide valuable insights to both the scientific community and policymakers, aiming to address the multifaceted challenges associated with emerging chemical contaminants in drinking water. Through our research, we endeavor to promote sustainable water treatment approaches and contribute to the advancement of strategies for safeguarding public health and environmental integrity.

2. Overall, the manuscript is well-written and organized. The introduction provides a clear background and rationale for the study, highlighting the importance of addressing emerging contaminants and their potential risks to human health and the environment. The methods section is detailed and comprehensive, allowing for reproducibility of the study. The use of the USEtox model for estimating health impacts and the life cycle assessment for evaluating environmental performance are appropriate and add credibility to the findings.

R: We sincerely appreciate the reviewer's positive feedback regarding the overall quality, organization, and clarity of our manuscript.

3. The results section presents the key findings in a clear and concise manner, supported by diverse figures and tables. The discussion section provides a thorough analysis and interpretation of the results, highlighting the implications and significance of the study. The conclusions are well-supported by the data and align with the objectives of the study.

R: We are grateful for the reviewer's commendation regarding the presentation of our key findings, the thorough analysis and interpretation of results in our discussion section, and the alignment of our conclusions with the study's objectives.

4. The study provides valuable insights into the assessment of emerging contaminants in drinking water and offers a comparative analysis of different water production

systems. It contributes to the existing literature in a meaningful way and will be of interest to researchers and practitioners in the field. I believe this manuscript is suitable for publication in our journal. However, there are a few minor revisions that I recommend addressing before publication.

R: We sincerely appreciate the reviewer's comprehensive assessment of our study and their recognition of its significance in providing insights into the modeling of emerging chemical contaminants in drinking water and the comparative analysis of different water purification systems from a water-environment-health nexus perspective. We have diligently addressed the minor revisions recommended by the reviewer to further enhance the quality and clarity of our manuscript. These revisions include careful consideration of the reviewer's feedback, ensuring that our work aligns with the high standards expected for publication in Nature Communications.

5. In the methods section, it would be helpful to provide additional information on the selection criteria for the included contaminants. Clarify if any specific criteria were used to prioritize the contaminants included in the analysis.

R: We have carefully considered this suggestion and have revised the methods section of our manuscript to offer a more detailed explanation of our criteria for selecting contaminants for inclusion in the analysis. Specifically, we clarify whether any specific criteria were employed to consider the contaminants and provide a rationale for including the chosen set of contaminants in the study (Main text, Page 5, Lines 101–105).

6. Provide more details on the uncertainty analysis conducted in the study. Describe the methodology used for the Monte Carlo analysis and how the uncertainty ranges were determined.

R: We acknowledge and accept this suggestion. In our revised manuscript, we have provided a detailed description of the methodology employed for the Monte Carlo analysis. This includes elucidating the process of generating random input parameter values and specifying the number of iterations conducted. Furthermore, we have expounded on how uncertainty ranges were determined for key model inputs, such as contaminant concentrations, toxicity factors, and removal efficiencies. We have also clarified the integration of these uncertainty ranges into the Monte Carlo analysis. Additionally, we have presented the results of the uncertainty analysis, including any sensitivity analyses undertaken to evaluate the impact of uncertainty on our findings. The relevant statements have been refined for enhanced clarity in the revised manuscript (Main text, Pages 26–27, Lines 628–662).

7. Consider discussing the limitations of the study in more detail. While the manuscript briefly mentions some limitations, a more comprehensive discussion in a separate paragraph would enhance the robustness of the study.

R: In response to the constructive feedback provided, we have revised the manuscript to include a separate paragraph devoted to discussing the potential constraints of our study (Main text, Pages 18–19, Lines 421–452). Within this dedicated section, we have provided a comprehensive overview of the limitations, encompassing both data and methodology, to ensure a thorough understanding. Firstly, we have addressed limitations associated with data sources, including considerations of data availability, quality, and potential sources of bias or uncertainty. This includes discussion of any limitations arising from data gaps, reliance on literature-based data, and the potential variability in contaminant occurrence across regions. Secondly, we have elaborated on methodological limitations, encompassing assumptions made in our modeling approach, simplifications, and any constraints associated with the models utilized. Furthermore, we have discussed the generalizability of our findings, highlighting the context-specific nature of our results and the extent to which they may apply to different geographic regions, water sources, or treatment systems. Lastly, we have utilized this paragraph to outline potential areas for future research, aiming to enhance our understanding of the water-environment-health nexus and refine the examination of water purification solutions. These revisions aim to provide readers with a comprehensive understanding of the limitations inherent in our study, while also identifying avenues for future research to advance this field.

8. The implications and practical applications of the findings could be further highlighted in the discussion section. Discuss how the results of this study can inform decision-making processes, water treatment strategies, and policy development.

R: We appreciate the reviewer’s suggestion to further emphasize the implications and practical applications of our findings in the discussion section. Acknowledging the importance of this aspect in providing valuable insights for decision-makers, water treatment professionals, and policymakers, we have expanded the discussion in our revised manuscript (Main text, Pages 20–21, Lines 480–501).

Response to Reviewer 5

1. The Authors answered to all my comments thoroughly and revised the paper. They added the paragraph: “Sensitivity analysis of disease burdens and environmental impacts” and extended considerably the discussion, however not always improving clarity. There remains an issue where the Authors' answer is not completely comprehensive, and it is the assessment of the uncertainty of the estimates. In fact, although the authors were comprehensive in stating all the elements of uncertainty in the estimates of DALYs, my comment referred to a quantitative assessment of the uncertainty in terms of the confidence intervals of the estimates. The Authors stated they used the Monte Carlo simulation to account for uncertainty but from my understanding by this technique they just incorporated variability.

R: We are grateful for the reviewer's feedback and acknowledge the need for further clarification regarding our approach to assessing uncertainty, specifically through confidence intervals, in our estimates of DALYs. We appreciate the opportunity to clarify our methodology and address any remaining concerns.

Firstly, we acknowledge the importance of distinguishing between variability and uncertainty in our analysis. While the Monte Carlo (MC) simulation was primarily used to capture variability among model parameters; it was also intended to address inherent uncertainties in the estimates. To clarify this distinction, we have included additional explanations regarding the role of the MC simulation in addressing both aspects, with particular emphasis on the extent to which model parameters influence the final estimates (*Main text, Page 7, Line 135; Page 19, Lines 458–462*). In response to the reviewer's request for a more quantitative assessment of uncertainty, we calculated confidence intervals for the DALY estimates to explicitly quantify this uncertainty. We have now included 95% confidence intervals alongside the DALY estimates in Figure 3 (*Main text, Page 34, Lines 870–872; Page 38, Figure 2*). To improve the clarity of the discussion on uncertainty, we have revised the relevant sections to explicitly outline the considerations made in assessing both variability and uncertainty (*Main text, Page 26, Lines 630–632, Lines 641–643*).

We hope these modifications address the reviewer's concerns and enhance the overall clarity of our manuscript. We appreciate the reviewer once again for the guidance on improving this crucial aspect of our work.

Response to Reviewer 7

1. I have thoroughly reviewed the manuscript and supplementary files as well as the authors' response to reviewers' letter. This is a detailed and highly relevant paper that contributes to the current literature in a meaningful way. This study examines the human-environment-water nexus by analyzing the presence, occurrence, and subsequent health impacts of contaminants of emerging concern in various water sources and determining the efficiency of different purification systems for drinking water. The manuscript is well-written, concise, and the authors appear to have taken a considerable amount of effort and time to address each of the reviewers' extensive comments in detail, while still remaining within scope of the originally proposed work. Methods are sound and presented in a manner that supports reproducibility. This study provides a foundation for future work to investigate more nuanced occurrences with regard to water quality parameters and impacts from CECs in various matrices, as noted in other comments by the other reviewers. I recommend this manuscript is fit for publication at this stage, with no further edits or comments to provide that would truly add to the presented work.

R: We would like to express our sincere gratitude to the reviewer for their thorough evaluation of our manuscript and for the encouraging comments. We are pleased to hear that the study's contribution to the field, the soundness of our methods, and our responses to prior comments were well received. We also appreciate for the reviewer's acknowledgement of the potential for this study to serve as a foundation for future investigations into water quality and health impacts from contaminants of emerging concern. We are sure that our work will contribute meaningfully to ongoing and future research in this vital area.

Response to Reviewer 8

1. While the authors have addressed the comments of the reviewers especially in terms of limitations of the study/modelling approach. However, the authors have not addressed the limitations of the study in the conclusion or abstract. It is imperative that conclusion clearly list out the limitations in making such conclusions. For example, better to soften the conclusion by incorporating phrases like "by analyzing two treatments trains". while I believe riverbank filtration is a strong candidate, discounting others or adding RO to this can be problematic, especially there can be instances where RBF or RBF-RO are not suitable. with the limitations of the study approach regarding the influences of the matrix of the water as well as influence of one process on subsequent processes and variability that can occur in various countries, the conclusion should still be softened. Your approach is novel but that does not mean you can conclude that it the only solution.

R: We appreciate the reviewer for this valuable feedback and for highlighting the need to emphasize the scope of our findings in both the abstract and conclusion. We understand that while our approach offers novel insights, it is important to clearly convey its scope and contextual limitations. In response to the reviewer's suggestions, we have made revisions in the manuscript.

Firstly, we have revised the Abstract to incorporate language that accurately reflects the implications of our findings. Specifically, we now utilize phrases such as "our analysis of two treatment alternatives" to clarify that our conclusions pertain solely to the scenarios we investigated (*Main text, Page 2, Lines 20–21*). Similarly, relevant conclusions in the Discussion section have been revised (*Main text, Page 13, Lines 300–301*). While our findings highlight the effectiveness of integrating RBF with RO in specific settings, these results should not be generalized across all water matrices or geographical locations. We explicitly acknowledge that other treatment approaches may be more suitable under different conditions (*Main text, Page 14, Lines 324–337; Page 15, Lines 344–350; Page 15–16, Lines 362–374*). Additionally, we have addressed limitations related to the influence of water matrices, interactions between treatment processes, and geographic variability (*Main text, Page 17, Lines 402–408*). Overall, we believe these revisions adequately address the reviewer's concern regarding the study's limitations and provide a more nuanced conclusion.

We hope these changes will meet with the reviewer's approval and enhance the manuscript's clarity and rigor.